# On the Value of Target Data in Transfer Learning

**Steve Hanneke**
Toyota Technological Institute at Chicago
`steve.hanneke@gmail.com`

**Samory Kpotufe**
Columbia University, Statistics
`skk2175@columbia.edu`

## Abstract

We aim to understand the value of additional labeled or unlabeled target data in transfer learning, for any given amount of source data; this is motivated by practical questions around minimizing sampling costs, whereby, target data is usually harder or costlier to acquire than source data, but can yield better accuracy.

To this aim, we establish the first minimax-rates in terms of both source and target sample sizes, and show that performance limits are captured by new notions of discrepancy between source and target, which we refer to as *transfer exponents*.

Interestingly, we find that attaining minimax performance is akin to ignoring one of the source or target samples, provided distributional parameters were known a priori. Moreover, we show that practical decisions – w.r.t. minimizing sampling costs – can be made in a minimax-optimal way *without* knowledge or estimation of distributional parameters nor of the discrepancy between source and target.

## 1   Introduction

The practice of transfer-learning often involves acquiring some amount of target data, and involves various practical decisions as to how to best combine source and target data; however much of the theoretical literature on transfer only addresses the setting where no target labeled data is available.

We aim to understand the value of target labels, that is, given $n_P$ labeled data from some source distribution $P$, and $n_Q$ labeled target labels from a target $Q$, what is the best $Q$ error achievable by any classifier in terms of *both* $n_Q$ and $n_P$, and which classifiers achieve such optimal transfer. In this first analysis, we mostly restrict ourselves to a setting, similar to the traditional *covariate-shift* assumption, where the best classifier – from a fixed VC class $\mathcal{H}$ – is the same under $P$ and $Q$.

We establish the first minimax-rates, for bounded-VC classes, in terms of both source and target sample sizes $n_P$ and $n_Q$, and show that performance limits are captured by new notions of discrepancy between source and target, which we refer to as *transfer exponents*.

The first notion of transfer-exponent, called $\rho$, is defined in terms of discrepancies in excess risk, and is most refined. Already here, our analysis reveals a surprising fact: the best possible rate (matching upper and lower-bounds) in terms of $\rho$ and both sample sizes $n_P, n_Q$ is - up to constants - achievable by an oracle which simply ignores the least informative of the source or target datasets. In other words, if $\hat{h}_P$ and $\hat{h}_Q$ denote the ERM on data from $P$, resp. from $Q$, one of the two achieves the optimal $Q$ rate over any classifier having access to both $P$ and $Q$ datasets. However, which of $\hat{h}_P$ or $\hat{h}_Q$ is optimal is not easily decided without prior knowledge: for instance, cross-validating on a holdout target-sample would naively result in a rate of $n_Q^{-1/2}$ which can be far from optimal given large $n_P$. Interestingly, we show that the optimal $(n_P, n_Q)$-rate is achieved by a generic approach, akin to so-called *hypothesis-transfer* [1, 2], which optimizes $Q$-error under the constraint of low $P$-error, and does so without knowledge of distributional parameters such as $\rho$.

We then consider a related notion of *marginal* transfer-exponent, called $\gamma$, defined w.r.t. marginals $P_X, Q_X$. This is motivated by the fact that practical decisions in transfer often involve the use of

cheaper unlabeled data (i.e., data drawn from $P_X, Q_X$). We will show that, when practical decisions are driven by observed changes in marginals $P_X, Q_X$, the marginal notion $\gamma$ is then most suited to capture performance as it does not require knowledge (or observations) of label distribution $Q_{Y|X}$.

In particular, the marginal exponent $\gamma$ helps capture performance limits in the following scenarios of current practical interest:

• **Minimizing sampling cost.** Given different costs of labeled source and target data, and a desired target excess error at most $\epsilon$, how to use unlabeled data to decide on an optimal sampling scheme that minimizes labeling costs while achieving target error at most $\epsilon$. (Section 6)

• **Choice of transfer.** Given two sources $P_1$ and $P_2$, each at some unknown distance from $Q$, given unlabeled data and some or no labeled data from $Q$, how to decide which of $P_1, P_2$ transfers best to the target $Q$. (Appendix A.2)

• **Reweighting.** Given some amount of unlabeled data from $Q$, and some or no labeled $Q$ data, how to optimally re-weight (out of a fixed set of schemes) the source $P$ data towards best target performance. While differently motivated, this problem is related to the last one. (Appendix A.1)

Although optimal decisions in the above scenarios depend tightly on unknown distributional parameters such as different label noise in source and target data, and on unknown *distance* from source to target (as captured by $\gamma$), we show that such practical decisions can be made, near optimally, with no knowledge of distributional parameters, and perhaps surprisingly, without ever estimating $\gamma$. Furthermore, the unlabeled sampling complexity can be shown to remain low. Finally, the procedures described in this work remain of a theoretical nature, but yield new insights into how various practical decisions in transfer can be made near-optimally in a data-driven fashion.

**Related Work.**   Much of the theoretical literature on transfer can be subdivided into a few main lines of work. As mentioned above, the main distinction with the present work is in that they mostly focus on situations with no labeled target data, and consider distinct notions of discrepancy between $P$ and $Q$. We contrast these various notions with the transfer-exponents $\rho$ and $\gamma$ in Section 3.1.

A first direction considers refinements of total-variation that quantify changes in error over classifiers in a fixed class $\mathcal{H}$. The most common such measures are the so-called $d_{\mathcal{A}}$-divergence [3, 4, 5] and the $\mathcal{Y}$-discrepancy [6, 7, 8]. In this line of work, the rates of transfer, largely expressed in terms of $n_P$ alone, take the form $o_p(1) + C \cdot \text{divergence}(P, Q)$. In other words, transfer down to $0$ error seems impossible whenever these divergences are non-negligible; we will carefully argue that such intuition can be overly pessimistic.

Another prominent line of work, which has led to many practical procedures, considers so-called density ratios $f_Q/f_P$ (importance weights) as a way to capture the similarity between $P$ and $Q$ [9, 10]. A related line of work considers information-theoretic measures such as KL-divergence or Renyi divergence [11, 12] but has received relatively less attention. Similar to these notions, the transfer-exponents $\rho$ and $\gamma$ are *asymmetric* measures of distance, attesting to the fact that it could be easier to transfer from some $P$ to $Q$ than the other way around. However, a significant downside to these notions is that they do not account for the specific structure of a hypothesis class $\mathcal{H}$ as is the case with the aforementionned divergences. As a result, they can be sensitive to issues such as minor differences of support in $P$ and $Q$, which may be irrelevant when learning with certain classes $\mathcal{H}$.

On the algorithmic side, many approaches assign importance weights to source data from $P$ so as to minimize some prescribed *metric* between $P$ and $Q$ [13, 14]; as we will argue, *metrics*, being symmetric, can be inadequate as a measure of discrepancy given the inherent asymmetry in transfer.

The importance of unlabeled data in transfer-learning, given the cost of target labels, has always been recognized, with various approaches developed over the years [15, 16], including more recent research efforts into so-called *semisupervised* or *active* transfer, where, given unlabeled target data, the goal is to request as few target labels as possible to improve classification over using source data alone [17, 18, 19, 20, 21].

More recently, [22, 23, 24] consider nonparametric transfer settings (unbounded VC) allowing for changes in conditional distributions. Also recent, but more closely related, [25] proposed a nonparametric measure of discrepancy which successfully captures the interaction between labeled source and target under nonparametric conditions and 0-1 loss; these notions however ignore the additional structure afforded by transfer in the context of a fixed hypothesis class.

## 2 Setup and Definitions

We consider a classification setting where the input $X \in \mathcal{X}$, some measurable space, and the output $Y \in \{0, 1\}$. We let $\mathcal{H} \subset 2^{\mathcal{X}}$ denote a fixed hypothesis class over $\mathcal{X}$, denote $d_{\mathcal{H}}$ the VC dimension [26], and the goal is to return a classifier $h \in \mathcal{H}$ with low error $R_Q(h) \doteq \mathbb{E}_Q[h(X) \neq Y]$ under some joint distribution $Q$ on $X, Y$. The learner has access to two independent labeled samples $S_P \sim P^{n_P}$ and $S_Q \sim Q^{n_Q}$, i.e., drawn from *source* distributions $P$ and target $Q$, of respective sizes $n_P, n_Q$. Our aim is to bound the excess error, under $Q$, of any $\hat{h}$ learned from both samples, in terms of $n_P, n_Q$, and (suitable) notions of discrepancy between $P$ and $Q$. We will let $P_X, Q_X, P_{Y|X}, Q_{Y|X}$ denote the corresponding marginal and conditional distributions under $P$ and $Q$.

**Definition 1.** For $D \in \{Q, P\}$, denote $\mathcal{E}_D(h) \doteq R_D(h) - \inf_{h' \in \mathcal{H}} R_D(h')$, the **excess error** of $h$.

**Distributional Conditions.** We consider various traditional assumptions in classification and transfer. The first one is a so-called *Bernstein Class Condition* on noise [27, 28, 29, 30, 31].

**(NC).** *Let $h_P^* \doteq \underset{h \in \mathcal{H}}{\operatorname{argmin}} R_P(h)$ and $h_Q^* \doteq \underset{h \in \mathcal{H}}{\operatorname{argmin}} R_Q(h)$ exist. $\exists \beta_P, \beta_Q \in [0, 1], c_P, c_Q > 0$ s.t.*

$$P_X(h \neq h_P^*) \leq c_p \cdot \mathcal{E}_P^{\beta_P}(h), \quad and \quad Q_X(h \neq h_Q^*) \leq c_q \cdot \mathcal{E}_Q^{\beta_Q}(h). \tag{1}$$

For instance, the usual Tsybakov noise condition, say on $P$, corresponds to the case where $h_P^*$ is the Bayes classifier, with corresponding regression function $\eta_P(x) \doteq \mathbb{E}[Y|x]$ satisfying $P_X(|\eta_P(X) - 1/2| \leq \tau) \leq C\tau^{\beta_P/(1-\beta_P)}$. Classification is easiest w.r.t. $P$ (or $Q$) when $\beta_P$ (resp. $\beta_Q$) is largest. We will see that this is also the case in Transfer.

The next assumption is stronger, but can be viewed as a relaxed version of the usual *Covariate-Shift* assumption which states that $P_{Y|X} = Q_{Y|X}$.

**(RCS).** *Let $h_P^*, h_Q^*$ as defined above. We have $\mathcal{E}_Q(h_P^*) = \mathcal{E}_Q(h_Q^*) = 0$. We then define $h^* \doteq h_P^*$.*

Note that the above allows $P_{Y|X} \neq Q_{Y|X}$. However, it is not strictly weaker than *Covariate-Shift*, since the latter allows $h_P^* \neq h_Q^*$ provided the Bayes $\notin \mathcal{H}$. The assumption is useful as it serves to isolate the sources of hardness in transfer beyond just shifts in $h^*$. We will in fact see later that it is easily removed, but at the additive (necessary) cost of $\mathcal{E}_Q(h_P^*)$.

## 3 Transfer-Exponents from $P$ to $Q$.

We consider various notions of *discrepancy* between $P$ and $Q$, which will be shown to tightly capture the complexity of transfer $P$ to $Q$.

**Definition 2.** We call $\rho > 0$ a **transfer-exponent** from $P$ to $Q$, w.r.t. $\mathcal{H}$, if there exists $C_\rho$ such that

$$\forall h \in \mathcal{H}, \quad C_\rho \cdot \mathcal{E}_P(h) \geq \mathcal{E}_Q^\rho(h). \tag{2}$$

We are interested in the smallest such $\rho$ with small $C_\rho$. We generally would think of $\rho$ as at least 1, although there are situations – which we refer to as *super-transfer*, to be discussed, where we have $\rho < 1$; in such situations, data from $P$ can yield faster $\mathcal{E}_Q$ rates than data from $Q$.

While the transfer-exponent will be seen to tightly capture the two-samples minimax rates of transfer, and can be *adapted to*, practical learning situations call for *marginal* versions that can capture the rates achievable when one has access to unlabeled $Q$ data.

**Definition 3.** We call $\gamma > 0$ a **marginal transfer-exponent** from $P$ to $Q$ if $\exists C_\gamma$ such that

$$\forall h \in \mathcal{H}, \quad C_\gamma \cdot P_X(h \neq h_P^*) \geq Q_X^\gamma(h \neq h_P^*). \tag{3}$$

The following simple proposition relates $\gamma$ to $\rho$.

**Proposition 1** (From $\gamma$ to $\rho$)**.** *Suppose Assumptions (NC) and (RCS) hold, and that $P$ has marginal transfer-exponent $(\gamma, C_\gamma)$ w.r.t. $Q$. Then $P$ has transfer-exponent $\rho \leq \gamma/\beta_P$, where $C_\rho = C_\gamma^{\gamma/\beta_P}$.*

*Proof.* $\forall h \in \mathcal{H}$, we have $\mathcal{E}_Q(h) \leq Q_X(h \neq h_P^*) \leq C_\gamma \cdot P_X(h \neq h_P^*)^{1/\gamma} \leq C_\gamma \cdot \mathcal{E}_P(h)^{\beta_P/\gamma}$. $\square$

## 3.1 Examples and Relation to other notions of discrepancy.

In this section, we consider various examples that highlight interesting aspects of $\rho$ and $\gamma$, and their relations to other notions of distance $P \to Q$ considered in the literature. Though our results cover noisy cases, in all these examples we assume no noise for simplicity, and therefore $\gamma = \rho$.

**Example 1.** (Non-overlapping supports) This first example emphasizes the fact that, unlike in much of previous analyses of transfer, the exponents $\gamma, \rho$ do not require that $Q_X$ and $P_X$ have overlapping support. This is a welcome property shared also by the $d_A$ and $\mathcal{Y}$ discrepancy.

In the example shown on the right, $\mathcal{H}$ is the class of homogeneous linear separators, while $P_X$ and $Q_X$ are uniform on the surface of the spheres depicted (e.g., corresponding to different scalings of the data). We then have that $\gamma = \rho = 1$ with $C_\gamma = 1$, while notions such as *density-ratios*, KL-divergences, or the recent nonparameteric notion of [25], are ill-defined or diverge to $\infty$.

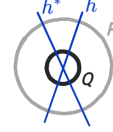

**Example 2.** (Large $d_A, d_{\mathcal{Y}}$) Let $\mathcal{H}$ be the class of one-sided thresholds on the line, but now we let $P_X \doteq \mathcal{U}[0, 2]$ and $Q_X \doteq \mathcal{U}[0, 1]$. Let $h^*$ be thresholded at $1/2$. We then see that for all $h_t$ thresholded at $t \in [0, 1]$, $2P_X(h_t \neq h^*) = \frac{1}{2}Q_X(h_t \neq h^*)$, where for $t > 1$, $P_X(h_t \neq h^*) = \frac{1}{2}(t - 1/2) \geq \frac{1}{2}Q_X(h_t \neq h^*) = \frac{1}{4}$. Thus, the marginal transfer exponent $\gamma = 1$ with $C_\gamma = 2$, so we have fast transfer at the same rate $1/n_P$ as if we were sampling from $Q$ (Theorem 3).

On the other hand, recall that the $d_A$-divergence takes the form $d_A(P, Q) \doteq \sup_{h \in \mathcal{H}} |P_X(h \neq h^*) - Q_X(h \neq h^*)|$, while the $\mathcal{Y}$-discrepancy takes the form $d_{\mathcal{Y}}(P, Q) \doteq \sup_{h \in \mathcal{H}} |\mathcal{E}_P(h) - \mathcal{E}_Q(h)|$. The two coincide whenever there is no noise in $Y$.

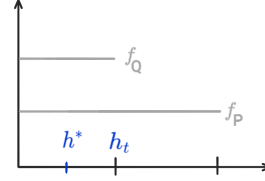

Now, take $h_t$ as the threshold at $t = 1/2$, and $d_A = d_{\mathcal{Y}} = \frac{1}{4}$ which would wrongly imply that transfer is not feasible at a rate faster than $\frac{1}{4}$; we can in fact make this situation worse, i.e., let $d_A = d_{\mathcal{Y}} \to \frac{1}{2}$ by letting $h^*$ correspond to a threshold close to 0. A first issue is that these divergences get large in large disagreement regions; this is somewhat mitigated by *localization*, as discussed in Example 4.

**Example 3.** (Minimum $\gamma$, $\rho$, and the inherent **asymmetry** of transfer) Suppose $\mathcal{H}$ is the class of one-sided thresholds on the line, $h^* = h_P^* = h_Q^*$ is a threshold at 0. The marginal $Q_X$ has uniform density $f_Q$ (on an interval containing 0), while, for some $\gamma \geq 1$, $P_X$ has density $f_P(t) \propto t^{\gamma-1}$ on $t > 0$ (and uniform on the rest of the support of $Q$, not shown). Consider any $h_t$ at threshold $t > 0$, we have $P_X(h_t \neq h^*) = \int_0^t f_P \propto t^\gamma$, while $Q_X(h_t \neq h^*) \propto t$. Notice that for any fixed $\epsilon > 0$,

$$\lim_{t>0,\, t\to0} \frac{Q_X(h_t \neq h^*)^{\gamma-\epsilon}}{P_X(h_t \neq h^*)} = \lim_{t>0,\, t\to0} C \frac{t^{\gamma-\epsilon}}{t^\gamma} = \infty.$$

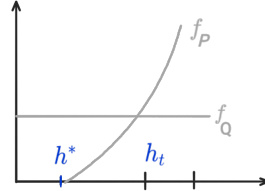

We therefore see that $\gamma$ is the smallest possible marginal transfer-exponent (similarly, $\rho = \gamma$ is the smallest possible transfer exponent). Interestingly, now consider transferring instead from $Q$ to $P$: we would have $\gamma(Q \to P) = 1 \leq \gamma \doteq \gamma(P \to Q)$, i.e., it could be easier to transfer from $Q$ to $P$ than from $P$ to $Q$, which is not captured by symmetric notions of distance ($d_A$, Wassertein, etc ...).

Finally note that the above example can be extended to more general hypothesis classes as it simply plays on how fast $f_P$ decreases w.r.t. $f_Q$ in regions of space.

**Example 4.** (*Super-transfer* and localization). We continue on the above Example 2. Now let $0 < \gamma < 1$, and let $f_P(t) \propto |t|^{\gamma-1}$ on $[-1, 1] \setminus \{0\}$, with $Q_X \doteq \mathcal{U}[-1, 1]$, $h^*$ at 0. As before, $\gamma$ is a transfer-exponent $P \to Q$, and following from Theorem 3, we attain transfer rates of $\mathcal{E}_Q \lesssim n_P^{-1/\gamma}$, faster than the rates of $n_Q^{-1}$ attainable with data from $Q$. We call these situations *super-transfer*, i.e., ones where the source data gets us faster to $h^*$; here $P$ concentrates more mass close to $h^*$, while more generally, such situations can also be constructed by letting $P_{Y|X}$ be less noisy than $Q_{Y|X}$ data, for instance corresponding to controlled lab data as source, vs noisy real-world data.

Now consider the following $\epsilon$-*localization* fix to the $d_A = d_{\mathcal{Y}}$ divergences over $h$'s with small $P$ error (assuming we only observe data from $P$): $d_{\mathcal{Y}}^* \doteq \sup_{h \in \mathcal{H}:\, \mathcal{E}_P(h) \leq \epsilon} |\mathcal{E}_P(h) - \mathcal{E}_Q(h)|$. This is no longer worst-case over all $h$'s, yet it is still not a complete fix. To see why, consider that, given $n_P$ data from $P$, the best $P$-excess risk attainable is $n_P^{-1}$ so we might set $\epsilon \propto n_P^{-1}$. Now the subclass $\{h \in \mathcal{H} : \mathcal{E}_P(h) \leq \epsilon\}$ corresponds to thresholds $t \in [\pm n_P^{-1/\gamma}]$, since $\mathcal{E}_P(h_t) = P([0, t]) \propto |t|^\gamma$.

We therefore have $d_{\mathcal{Y}}^* \propto \left| n_P^{-1} - n_P^{-1/\gamma} \right| \propto n_P^{-1}$, wrongly suggesting a transfer rate $\mathcal{E}_Q \lesssim n_P^{-1}$, while the super-transfer rate $n_P^{-1/\gamma}$ is achievable as discussed above. The problem is that, even after localization, $d_{\mathcal{Y}}^*$ treats errors under $P$ and $Q$ symmetrically.

## 4 Lower-Bounds

**Definition 4** ((NC) Class). *Let $\mathcal{F}_{(NC)}(\rho, \beta_P, \beta_Q, C)$ denote the class of pairs of distributions $(P, Q)$ with transfer-exponent $\rho$, $C_\rho \leq C$, satisfying (NC) with parameters $\beta_P, \beta_Q$, and $c_P, c_Q \leq C$.*

The following lower-bound in terms of $\rho$ is obtained via information theoretic-arguments. In effect, given the VC class $\mathcal{H}$, we construct a set of distribution pairs $\{(P_i, Q_i)\}$ supported on $d_{\mathcal{H}}$ datapoints, which all belong to $\mathcal{F}_{(NC)}(\rho, \beta_P, \beta_Q, C)$. All the distributions share the same marginals $P_X, Q_X$. Any two pairs are close to each other in the sense that $\Pi_i, \Pi_j$, where $\Pi_i \doteq P_i^{n_P} \times Q_i^{n_Q}$, are close in KL-divergence, while, however maintaining pairs $(P_i, Q_i), (P_j, Q_j)$ far in a pseudo-distance induced by $Q_X$. All the proofs from this section are in Appendix B.

**Theorem 1** ($\rho$ Lower-bound). *Suppose the hypothesis class $\mathcal{H}$ has VC dimension $d_{\mathcal{H}} \geq 9$. Let $\hat{h} = \hat{h}(S_P, S_Q)$ denote any (possibly improper) classifier with access to two independent labeled samples $S_P \sim P^{n_P}$ and $S_Q \sim Q^{n_Q}$. Fix any $\rho \geq 1$, $0 \leq \beta_P, \beta_Q < 1$. Suppose either $n_P$ or $n_Q$ is sufficiently large so that*

$$\epsilon(n_P, n_Q) \doteq \min \left\{ \left( \frac{d_{\mathcal{H}}}{n_P} \right)^{1/(2-\beta_P)\rho}, \left( \frac{d_{\mathcal{H}}}{n_Q} \right)^{1/(2-\beta_Q)} \right\} \leq 1/2.$$

*Then, for any $\hat{h}$, there exists $(P, Q) \in \mathcal{F}_{(NC)}(\rho, \beta_P, \beta_Q, 1)$, and a universal constant $c$ such that,*

$$\mathbb{P}_{S_P, S_Q} \left( \mathcal{E}_Q(\hat{h}) > c \cdot \epsilon(n_P, n_Q) \right) \geq \frac{3 - 2\sqrt{2}}{8}.$$

As per Proposition 1 we can translate any upper-bound in terms of $\rho$ to an upper-bound in terms of $\gamma$ since $\rho \leq \gamma/\beta_P$. We investigate whether such upper-bounds in terms of $\gamma$ are tight, i.e., given a class $\mathcal{F}_{(NC)}(\rho, \beta_P, \beta_Q, C)$, are there distributions with $\rho = \gamma/\beta_P$ where the rate is realized.

The proof of the next result is similar to that of Theorem 1, however with the added difficulty that we need the construction to yield two forms of rates $\epsilon_1(n_P, n_Q), \epsilon_2(n_P, n_Q)$ over the data support (again $d_{\mathcal{H}}$ points). Combining these two rates matches the desired upper-bound. In effect, we follow the intuition that, to have $\rho = \gamma/\beta_P$ achieved on some subset $\mathcal{X}_1 \subset \mathcal{X}$, we need $\beta_Q$ to behave as 1 locally on $\mathcal{X}_1$, while matching the rate requires larger $\beta_Q$ on the rest of the suppport (on $\mathcal{X} \setminus \mathcal{X}_1$).

**Theorem 2** ($\gamma$ Lower-bound). *Suppose the hypothesis class $\mathcal{H}$ has VC dimension $d_{\mathcal{H}}$, $\lfloor d_{\mathcal{H}}/2 \rfloor \geq 9$. Let $\hat{h} = \hat{h}(S_P, S_Q)$ denote any (possibly improper) classifier with access to two independent labeled samples $S_P \sim P^{n_P}$ and $S_Q \sim Q^{n_Q}$. Fix any $0 < \beta_P, \beta_Q < 1$, $\rho \geq \max\{1/\beta_P, 1/\beta_Q\}$. Suppose either $n_P$ or $n_Q$ is sufficiently large so that*

$$\epsilon_1(n_P, n_Q) \doteq \min \left\{ \left( \frac{d_{\mathcal{H}}}{n_P} \right)^{1/(2-\beta_P)\rho \cdot \beta_Q}, \left( \frac{d_{\mathcal{H}}}{n_Q} \right)^{1/(2-\beta_Q)} \right\} \leq 1/2, \text{ and}$$

$$\epsilon_2(n_P, n_Q) \doteq \min \left\{ \left( \frac{d_{\mathcal{H}}}{n_P} \right)^{1/(2-\beta_P)\rho}, \left( \frac{d_{\mathcal{H}}}{n_Q} \right) \right\} \leq 1/2.$$

*Then, for any $\hat{h}$, there exists $(P, Q) \in \mathcal{F}_{(NC)}(\rho, \beta_P, \beta_Q, 2)$, with **marginal-transfer-exponent** $\gamma = \rho \cdot \beta_P \geq 1$, with $C_\gamma \leq 2$, and a universal constant $c$ such that,*

$$\mathbb{E}_{S_P, S_Q} \mathcal{E}_Q(\hat{h}) \geq c \cdot \max\{\epsilon_1(n_P, n_Q), \epsilon_2(n_p, n_Q)\}.$$

**Remark 1** (Tightness with upper-bound). *Write $\epsilon_1(n_P, n_Q) = \min\{\epsilon_1(n_P), \epsilon_1(n_Q)\}$, and similarly, $\epsilon_2(n_P, n_Q) = \min\{\epsilon_2(n_P), \epsilon_2(n_Q)\}$. Define $\epsilon_L \doteq \max\{\epsilon_1(n_P, n_Q), \epsilon_2(n_P, n_Q)\}$ as in the above lower-bound of Theorem 2. Next, define $\epsilon_H \doteq \min\{\epsilon_2(n_P), \epsilon_1(n_Q)\}$. It turns out that the*

*best upper-bound we can show (as a function of $\gamma$) is in terms of $\epsilon_H$ so defined. It is therefore natural to ask whether or when $\epsilon_H$ and $\epsilon_L$ are of the same order.*

*Clearly, we have $\epsilon_1(n_P) \leq \epsilon_2(n_P)$ and $\epsilon_1(n_Q) \geq \epsilon_2(n_Q)$ so that $\epsilon_L \leq \epsilon_H$ (as to be expected).*

*Now, if $\beta_Q = 1$, we have $\epsilon_1(n_P) = \epsilon_2(n_P)$ and $\epsilon_1(n_Q) = \epsilon_2(n_Q)$, so that $\epsilon_L = \epsilon_H$. More generally, from the above inequalities, we see that $\epsilon_L = \epsilon_H$ in the two regimes where either $\epsilon_1(n_Q) \leq \epsilon_1(n_P)$ (in which case $\epsilon_L = \epsilon_H = \epsilon_1(n_Q)$), or $\epsilon_2(n_P) \leq \epsilon_2(n_Q)$ (in which case $\epsilon_L = \epsilon_H = \epsilon_2(n_P)$).*

# 5 Upper-Bounds

The following lemma is due to [32].

**Lemma 1.** *Let $A_n = \frac{d_{\mathcal{H}}}{n} \log\left(\frac{\max\{n, d_{\mathcal{H}}\}}{d_{\mathcal{H}}}\right) + \frac{1}{n} \log\left(\frac{1}{\delta}\right)$. With probability at least $1 - \frac{\delta}{3}$, $\forall h, h' \in \mathcal{H}$,*

$$R(h) - R(h') \leq \hat{R}(h) - \hat{R}(h') + c\sqrt{\min\{P(h \neq h'), \hat{P}(h \neq h')\}A_n} + cA_n, \qquad (4)$$

*and*

$$\frac{1}{2}P(h \neq h') - cA_n \leq \hat{P}(h \neq h') \leq 2P(h \neq h') + cA_n, \qquad (5)$$

*for a universal numerical constant $c \in (0, \infty)$, where $\hat{R}$ denotes empirical risk on $n$ iid samples.*

Now consider the following algorithm. Let $S_P$ be a sequence of $n_P$ samples from $P$ and $S_Q$ a sequence of $n_Q$ samples from $Q$. Also let $\hat{h}_{S_P} = \operatorname{argmin}_{h \in \mathcal{H}} \hat{R}_{S_P}(h)$ and $\hat{h}_{S_Q} = \operatorname{argmin}_{h \in \mathcal{H}} \hat{R}_{S_Q}(h)$. Choose $\hat{h}$ as the solution to the following optimization problem.

---

Algorithm 1:

    Minimize    $\hat{R}_{S_P}(h)$

    subject to    $\hat{R}_{S_Q}(h) - \hat{R}_{S_Q}(\hat{h}_{S_Q}) \leq c\sqrt{\hat{P}_{S_Q}(h \neq \hat{h}_{S_Q})A_{n_Q}} + cA_{n_Q}$    (6)

               $h \in \mathcal{H}.$

---

The intuition is that, effectively, the constraint guarantees we maintain a near-optimal guarantee on $\mathcal{E}_Q(\hat{h})$ in terms of $n_Q$ and the (NC) parameters for $Q$, while (as we show) still allowing the algorithm to select an $h$ with a near-minimal value of $\hat{R}_{S_P}(h)$. The latter guarantee plugs into the transfer condition to obtain a term converging in $n_P$, while the former provides a term converging in $n_Q$, and altogether the procedure achieves a rate specified by the *min* of these two guarantees (which is in fact nearly minimax *optimal*, since it matches the lower bound up to logarithmic factors).

Formally, we have the following result for this learning rule; its proof is below.

**Theorem 3** (Minimax Upper-Bounds). *Assume (NC). Let $\hat{h}$ be the solution from Algorithm 1. For a constant $C$ depending on $\rho, C_\rho, \beta_P, c_{\beta_P}, \beta_Q, c_{\beta_Q}$, with probability at least $1 - \delta$,*

$$\mathcal{E}_Q(\hat{h}) \leq C \min\left\{A_{n_P}^{\frac{1}{(2-\beta_P)\rho}}, A_{n_Q}^{\frac{1}{2-\beta_Q}}\right\} = \tilde{O}\left(\min\left\{\left(\frac{d_{\mathcal{H}}}{n_P}\right)^{\frac{1}{(2-\beta_P)\rho}}, \left(\frac{d_{\mathcal{H}}}{n_Q}\right)^{\frac{1}{2-\beta_Q}}\right\}\right).$$

Note that, by the lower bound of Theorem 1, this bound is optimal up to log factors.

**Remark 2** (Effective Source Sample Size). *From the above, we might view (ignoring $d_{\mathcal{H}}$) $\tilde{n}_P \doteq n_P^{(2-\beta_Q)/(2-\beta_P)\rho}$ as the effective sample size contributed by $P$. In fact, the above minimax rate is of order $(\tilde{n}_P + n_Q)^{-1/(2-\beta_Q)}$, which yields added intuition into the combined effect of both samples. We have that, the effective source sample size $\tilde{n}_P$ is smallest for large $\rho$, but also depends on $(2-\beta_Q)/(2-\beta_P)$, i.e., on whether $P$ is noisier than $Q$.*

**Remark 3** (Rate in terms of $\gamma$). *Note that, by Proposition 1, this also immediately implies a bound under the marginal transfer condition and RCS, simply taking $\rho \leq \gamma/\beta_P$. Furthermore, by the lower bound of Theorem 2, the resulting bound in terms of $\gamma$ is tight in certain regimes up to log factors.*

*Proof of Theorem 3.* In all the lines below, we let $C$ serve as a generic constant (possibly depending on $\rho, C_\rho, \beta_P, c_{\beta_P}, \beta_Q, c_{\beta_Q}$) which may be different in different appearances. Consider the event of probability at least $1 - \delta/3$ from Lemma 1 for the $S_Q$ samples. In particular, on this event, if $\mathcal{E}_Q(h_P^*) = 0$, it holds that

$$\hat{R}_{S_Q}(h_P^*) - \hat{R}_{S_Q}(\hat{h}_{S_Q}) \leq c\sqrt{\hat{P}_{S_Q}(h_P^* \neq \hat{h}_{S_Q})A_{n_Q}} + cA_{n_Q}.$$

This means, under the (RCS) condition, $h_P^*$ satisfies the constraint in the above optimization problem defining $\hat{h}$. Also, on this same event from Lemma 1 we have

$$\mathcal{E}_Q(\hat{h}_{S_Q}) \leq c\sqrt{Q(\hat{h}_{S_Q} \neq h_Q^*)A_{n_Q}} + cA_{n_Q},$$

so that (NC) implies

$$\mathcal{E}_Q(\hat{h}_{S_Q}) \leq C\sqrt{\mathcal{E}_Q(\hat{h}_{S_Q})^{\beta_Q}A_{n_Q}} + cA_{n_Q},$$

which implies the well-known fact from [28, 29] that

$$\mathcal{E}_Q(\hat{h}_{S_Q}) \leq C\left(\frac{d_{\mathcal{H}}}{n_Q}\log\left(\frac{n_Q}{d_{\mathcal{H}}}\right) + \frac{1}{n_Q}\log\left(\frac{1}{\delta}\right)\right)^{\frac{1}{2-\beta_Q}}. \tag{7}$$

Furthermore, following the analogous argument for $S_P$, it follows that for any set $\mathcal{G} \subseteq \mathcal{H}$ with $h_P^* \in \mathcal{G}$, with probability at least $1 - \delta/3$, the ERM $\hat{h}'_{S_P} = \operatorname{argmin}_{h \in \mathcal{G}} \hat{R}_{S_P}(h)$ satisfies

$$\mathcal{E}_P(\hat{h}'_{S_P}) \leq C\left(\frac{d_{\mathcal{H}}}{n_P}\log\left(\frac{n_P}{d_{\mathcal{H}}}\right) + \frac{1}{n_P}\log\left(\frac{1}{\delta}\right)\right)^{\frac{1}{2-\beta_P}}. \tag{8}$$

In particular, conditioned on the $S_Q$ data, we can take the set $\mathcal{G}$ as the set of $h \in \mathcal{H}$ satisfying the constraint in the optimization, and on the above event we have $h_P^* \in \mathcal{G}$ (assuming the (RCS) condition); furthermore, if $\mathcal{E}_Q(h_P^*) = 0$, then without loss we can simply define $h_Q^* = h_P^* = h^*$ (and it is easy to see that this does not affect the NC condition). We thereby establish the above inequality (8) for this choice of $\mathcal{G}$, in which case by definition $\hat{h}'_{S_P} = \hat{h}$. Altogether, by the union bound, all of these events hold simultaneously with probability at least $1 - \delta$. In particular, on this event, if the (RCS) condition holds then

$$\mathcal{E}_P(\hat{h}) \leq C\left(\frac{d_{\mathcal{H}}}{n_P}\log\left(\frac{n_P}{d_{\mathcal{H}}}\right) + \frac{1}{n_P}\log\left(\frac{1}{\delta}\right)\right)^{\frac{1}{2-\beta_P}}.$$

Applying the definition of $\rho$, this has the further implication that (again if (RCS) holds)

$$\mathcal{E}_Q(\hat{h}) \leq C\left(\frac{d_{\mathcal{H}}}{n_P}\log\left(\frac{n_P}{d_{\mathcal{H}}}\right) + \frac{1}{n_P}\log\left(\frac{1}{\delta}\right)\right)^{\frac{1}{(2-\beta_P)\rho}}.$$

Also note that, if $\rho = \infty$ this inequality trivially holds, whereas if $\rho < \infty$ then (RCS) necessarily holds so that the above implication is generally valid, without needing the (RCS) assumption explicitly. Moreover, again when the above events hold, using the event from Lemma 1 again, along with the constraint from the optimization, we have that

$$R_Q(\hat{h}) - R_Q(\hat{h}_{S_Q}) \leq 2c\sqrt{\hat{P}_{S_Q}(\hat{h} \neq \hat{h}_{S_Q})A_{n_Q}} + 2cA_{n_Q},$$

and (5) implies the right hand side is at most

$$C\sqrt{Q(\hat{h} \neq \hat{h}_{S_Q})A_{n_Q}} + CA_{n_Q} \leq C\sqrt{Q(\hat{h} \neq h_Q^*)A_{n_Q}} + C\sqrt{Q(\hat{h}_{S_Q} \neq h_Q^*)A_{n_Q}} + CA_{n_Q}.$$

Using the Bernstein class condition and (7), the second term is bounded by

$$C\left(\frac{d_{\mathcal{H}}}{n_Q}\log\left(\frac{n_Q}{d_{\mathcal{H}}}\right) + \frac{1}{n_Q}\log\left(\frac{1}{\delta}\right)\right)^{\frac{1}{2-\beta_Q}},$$

while the first term is bounded by

$$C\sqrt{\mathcal{E}_Q(\hat{h})^{\beta_Q}A_{n_Q}}.$$

Altogether, we have that

$$\mathcal{E}_Q(\hat{h}) = R_Q(\hat{h}) - R_Q(\hat{h}_{S_Q}) + \mathcal{E}_Q(\hat{h}_{S_Q})$$

$$\leq C\sqrt{\mathcal{E}_Q(\hat{h})^{\beta_Q} A_{n_Q}} + C\left(\frac{d_{\mathcal{H}}}{n_Q}\log\left(\frac{n_Q}{d_{\mathcal{H}}}\right) + \frac{1}{n_Q}\log\left(\frac{1}{\delta}\right)\right)^{\frac{1}{2-\beta_Q}},$$

which implies

$$\mathcal{E}_Q(\hat{h}) \leq C\left(\frac{d_{\mathcal{H}}}{n_Q}\log\left(\frac{n_Q}{d_{\mathcal{H}}}\right) + \frac{1}{n_Q}\log\left(\frac{1}{\delta}\right)\right)^{\frac{1}{2-\beta_Q}}. \qquad \square$$

**Remark 4.** *Note that the above Theorem 3 does not require (RCS): that is, it holds even when* $\mathcal{E}_Q(h_P^*) > 0$, *in which case* $\rho = \infty$. *However, for a related method we can also show a stronger result in terms of a modified definition of* $\rho$:

Specifically, define $\mathcal{E}_Q(h, h_P^*) = \max\{R_Q(h) - R_Q(h_P^*), 0\}$, and suppose $\rho' > 0$, $C_{\rho'} > 0$ satisfy

$$\forall h \in \mathcal{H}, \quad C_{\rho'} \cdot \mathcal{E}_P(h) \geq \mathcal{E}_Q^{\rho'}(h, h_P^*).$$

This is clearly equivalent to $\rho$ (Definition 2) under (RCS); however, unlike $\rho$, this $\rho'$ can be finite even in cases where (RCS) fails. With this definition, we have the following result.

**Proposition 2** (Beyond (RCS)). *If* $\hat{R}_{S_Q}(\hat{h}_{S_P}) - \hat{R}_{S_Q}(\hat{h}_{S_Q}) \leq c\sqrt{\hat{P}_{S_Q}(\hat{h}_{S_P} \neq \hat{h}_{S_Q})A_{n_Q}} + cA_{n_Q}$, *that is, if* $\hat{h}_{S_P}$ *satisfies (6), define* $\hat{h} = \hat{h}_{S_P}$, *and otherwise define* $\hat{h} = \hat{h}_{S_Q}$. *Assume (NC). For a constant* $C$ *depending on* $\rho', C_{\rho'}, \beta_P, c_{\beta_P}, \beta_Q, c_{\beta_Q}$, *with probability at least* $1 - \delta$,

$$\mathcal{E}_Q(\hat{h}) \leq \min\left\{\mathcal{E}_Q(h_P^*) + CA_{n_P}^{\frac{1}{(2-\beta_P)\rho'}}, CA_{n_Q}^{\frac{1}{2-\beta_Q}}\right\}.$$

The proof of this result is similar to that of Theorem 3, and as such is deferred to Appendix C.

**An alternative procedure.** Similar results as in Theorem 3 can be obtained for a method that swaps the roles of $P$ and $Q$ samples:

---

Algorithm 1′ :

    Minimize       $\hat{R}_{S_Q}(h)$

    subject to     $\hat{R}_{S_P}(h) - \hat{R}_{S_P}(\hat{h}_{S_P}) \leq c\sqrt{\hat{P}_{S_P}(h \neq \hat{h}_{S_P})A_{n_P}} + cA_{n_P}$

                $h \in \mathcal{H}.$

---

This version, more akin to so-called *hypothesis transfer* may have practical benefits in scenarios where the $P$ data is accessible *before* the $Q$ data, since then the feasible set might be calculated (or approximated) in advance, so that the $P$ data itself would no longer be needed in order to execute the procedure. However this procedure presumes that $h_P^*$ is not far from $h_Q^*$, i.e., that data $S_P$ from $P$ is not misleading, since it conditions on doing well on $S_P$. Hence we now require (RCS).

**Proposition 3.** *Assume (NC) and (RCS). Let* $\hat{h}$ *be the solution from Algorithm 1′. For a constant* $C$ *depending on* $\rho, C_\rho, \beta_P, c_{\beta_P}, \beta_Q, c_{\beta_Q}$, *with probability at least* $1 - \delta$,

$$\mathcal{E}_Q(\hat{h}) \leq C\min\left\{A_{n_P}^{\frac{1}{(2-\beta_P)\rho}}, A_{n_Q}^{\frac{1}{2-\beta_Q}}\right\} = \tilde{O}\left(\min\left\{\left(\frac{d_{\mathcal{H}}}{n_P}\right)^{\frac{1}{(2-\beta_P)\rho}}, \left(\frac{d_{\mathcal{H}}}{n_Q}\right)^{\frac{1}{2-\beta_Q}}\right\}\right).$$

The proof is very similar to that of Theorem 3, so is omitted for brevity.

## 6 Minimizing Sampling Cost

In this section (and continued in Appendix A.1), we discuss the value of having access to *unlabeled* data from $Q$. The idea is that unlabeled data can be obtained much more cheaply than labeled data,

so gaining access to unlabeled data can be realistic in many applications. Specifically, we begin by discussing an adaptive sampling scenario, where we are able to *draw* samples from $P$ or $Q$, at different *costs*, and we are interested in optimizing the total cost of obtaining a given excess $Q$-risk.

Formally, consider the scenario where we have as input a value $\epsilon$, and are tasked with producing a classifier $\hat{h}$ with $\mathcal{E}_Q(\hat{h}) \leq \epsilon$. We are then allowed to *draw* samples from either $P$ or $Q$ toward achieving this goal, but at different costs. Suppose $\mathfrak{c}_P : \mathbb{N} \to [0, \infty)$ and $\mathfrak{c}_Q : \mathbb{N} \to [0, \infty)$ are *cost functions*, where $\mathfrak{c}_P(n)$ indicates the cost of sampling a batch of size $n$ from $P$, and similarly define $\mathfrak{c}_Q(n)$. We suppose these functions are increasing, and concave, and unbounded.

**Definition 5.** Define $n_Q^* = d_{\mathcal{H}}/\epsilon^{2-\beta_Q}$, $n_P^* = d_{\mathcal{H}}/\epsilon^{(2-\beta_P)\gamma/\beta_P}$, and $\mathfrak{c}^* = \min\{\mathfrak{c}_Q(n_Q^*), \mathfrak{c}_P(n_P^*)\}$. We call $\mathfrak{c}^* = \mathfrak{c}^*(\epsilon; \mathfrak{c}_P, \mathfrak{c}_Q)$ the **minimax optimal cost** of sampling from $P$ or $Q$ to attain $Q$-error $\epsilon$.

Note that the cost $\mathfrak{c}^*$ is effectively the smallest possible, up to log factors, in the range of parameters given in Theorem 2. That is, in order to make the lower bound in Theorem 2 less than $\epsilon$, either $n_Q = \tilde{\Omega}(n_Q^*)$ samples are needed from $Q$ or $n_P = \tilde{\Omega}(n_P^*)$ samples are needed from $P$. We show that $\mathfrak{c}^*$ is nearly achievable, adaptively with no knowledge of distributional parameters.

**Procedure.** We assume access to a large unlabeled data set $U_Q$ sampled from $Q_X$. For our purposes, we will suppose this data set has size at least $\Theta(\frac{d_{\mathcal{H}}}{\epsilon}\log\frac{1}{\epsilon} + \frac{1}{\epsilon}\log\frac{1}{\delta})$.

Let $A'_n = \frac{d_{\mathcal{H}}}{n}\log(\frac{\max\{n,d_{\mathcal{H}}\}}{d_{\mathcal{H}}}) + \frac{1}{n}\log(\frac{2n^2}{\delta})$. Then for any labeled data set $S$, define $\hat{h}_S = \operatorname{argmin}_{h \in \mathcal{H}} \hat{R}_S(h)$, and given an additional data set $U$ (labeled or unlabeled) define a quantity

$$\hat{\delta}(S, U) = \sup\left\{\hat{P}_U(h \neq \hat{h}_S) : h \in \mathcal{H}, \hat{R}_S(h) - \hat{R}_S(\hat{h}_S) \leq c\sqrt{\hat{P}_S(h \neq \hat{h}_S)A'_{|S|}} + cA'_{|S|}\right\},$$

where $c$ is as in Lemma 1. Now we have the following procedure.

---

Algorithm 2:
0. $S_P \leftarrow \{\}$, $S_Q \leftarrow \{\}$
1. For $t = 1, 2, \ldots$
2.     Let $n_{t,P}$ be minimal such that $\mathfrak{c}_P(n_{t,P}) \geq 2^{t-1}$
3.     Sample $n_{t,P}$ samples from $P$ and add them to $S_P$
4.     Let $n_{t,Q}$ be minimal such that $\mathfrak{c}_Q(n_{t,Q}) \geq 2^{t-1}$
5.     Sample $n_{t,Q}$ samples from $Q$ and add them to $S_Q$
6.     If $c\sqrt{\hat{\delta}(S_Q, S_Q)A_{|S_Q|}} + cA_{|S_Q|} \leq \epsilon$, return $\hat{h}_{S_Q}$
7.     If $\hat{\delta}(S_P, U_Q) \leq \epsilon/4$, return $\hat{h}_{S_P}$

---

The following theorem asserts that this procedure will find a classifier $\hat{h}$ with $\mathcal{E}_Q(\hat{h}) \leq \epsilon$ while adaptively using a near-minimal cost associated with achieving this. The proof is in Appendix D.

**Theorem 4** (Adapting to Sampling Costs). *Assume (NC) and (RCS). There exist a constant $c'$, depending on parameters ($C_\gamma$, $\gamma$, $c_{\beta_Q}$, $\beta_Q$, $c_{\beta_P}$, $\beta_P$) but not on $\epsilon$ or $\delta$, such that the following holds. Define sample sizes $\tilde{n}_Q = \frac{c'}{\epsilon^{2-\beta_Q}}\left(d_{\mathcal{H}}\log\frac{1}{\epsilon} + \log\frac{1}{\delta}\right)$, and $\tilde{n}_P = \frac{c'}{\epsilon^{(2-\beta_P)\gamma/\beta_P}}\left(d_{\mathcal{H}}\log\frac{1}{\epsilon} + \log\frac{1}{\delta}\right)$.*

*Algorithm 2 outputs a classifier $\hat{h}$ such that, with probability at least $1 - \delta$, we have $\mathcal{E}_Q(\hat{h}) \leq \epsilon$, and the total sampling cost incurred is at most $\min\{\mathfrak{c}_Q(\tilde{n}_Q), \mathfrak{c}_P(\tilde{n}_P)\} = \tilde{O}(\mathfrak{c}^*)$.*

Thus, when $\mathfrak{c}^*$ favors sampling from $P$, we end up sampling very few labeled $Q$ data. These are scenarios where $P$ samples are cheap relative to the cost of $Q$ samples and w.r.t. parameters ($\beta_Q, \beta_P, \gamma$) which determine the effective source sample size contributed for every target sample. Furthermore, we achieve this adaptively: without knowing (or even estimating) these relevant parameters.

**Acknowledgments**

We thank Mehryar Mohri for several very important discussions which helped crystallize many essential questions and directions on this topic.

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
