[Supplementary Material · ERM_Transfer.pdf]

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

## A  Additional Results

### A.1  Reweighting the Source Data

In this section, we present a technique for using unlabeled data from $Q$ to find a reweighting of the $P$ data more suitable for transfer. This gives a technique for using the data effectively in a potentially practical way. As above, we again suppose access to the sample $U_Q$ of unlabeled data from $Q$.

Additionally, we suppose we have access to a set $\mathcal{P}$ of functions $f : \mathcal{X} \to [0, \infty)$, which we interpret as unnormalized *density* functions with respect to $P_X$. Let $P_f$ denote the bounded measure whose marginal on $\mathcal{X}$ has density $f$ with respect to $P_X$, and the conditional $Y|X$ is the same as for $P$.

Now suppose $S_P = \{(x_i, y_i)\}_{i=1}^{n_P}$ is a sequence of $n_P$ iid $P$-distributed samples. Continuing conventions from above $R_{P_f}(h) = \int \mathbb{1}[h(x) \neq y] f(x) \mathrm{d}P(x, y)$ is a risk with respect to $P_f$, but now we also write $\hat{R}_{S_P, f}(h) = \frac{1}{n_P} \sum_{(x,y) \in S_P} \mathbb{1}[h(x) \neq y] f(x)$, and additionally we will use $P_{f^2}(h \neq h') = \int \mathbb{1}[h(x) \neq h'(x)] f^2(x) \mathrm{d}P(x, y)$, and $\hat{P}_{S_P, f^2}(h \neq h') = \frac{1}{n_P} \sum_{(x,y) \in S_P} \mathbb{1}[h(x) \neq h'(x)] f^2(x)$; the reason $f^2$ is used instead of $f$ is that this will represent a variance term in the bounds below. Other notations from above are defined analogously. In particular, also let $\hat{h}_{S_P, f} = \operatorname{argmin}_{h \in \mathcal{H}} \hat{R}_{S_P, f}(h)$. For simplicity, we will only present the case of $\mathcal{P}$ having finite pseudo-dimension $d_p$ (i.e., $d_p$ is the VC dimension of the subgraph functions $\{(x, y) \mapsto \mathbb{1}[f(x) \leq y] : f \in \mathcal{P}\}$); extensions to general bracketing or empirical covering follow similarly.

For the remaining results in this section, we suppose the condition RCS holds for *all* $P_f$: that is, $R_{P_f}$ is minimized in $\mathcal{H}$ at a function $h_{P_f}^*$ having $\mathcal{E}_Q(h_{P_f}^*) = 0$. For instance, this would be the case if the Bayes optimal classifier is in the class $\mathcal{H}$.

Define $A_n'' = \frac{d_{\mathcal{H}} + d_p}{n} \log\left(\frac{\max\{n, d_{\mathcal{H}} + d_p\}}{d_{\mathcal{H}} + d_p}\right) + \frac{1}{n} \log\left(\frac{1}{\delta}\right)$. Let us also extend the definition of $\hat{\delta}$ introduced above. Specifically, define $\hat{\delta}(S_P, f, U_Q)$ as

$$\sup\left\{\hat{P}_{U_Q}(h \neq \hat{h}_{S_P, f}) : h \in \mathcal{H}, \hat{\mathcal{E}}_{S_P, f}(h) \leq c\sqrt{\hat{P}_{S_P, f^2}(h \neq \hat{h}_{S_P, f}) A_{n_P}''} + c\|f\|_\infty A_{n_P}''\right\}.$$

Now consider the following procedure.

---
Algorithm 3:
Choose $\hat{f}$ to minimize $\hat{\delta}(S_P, f, U_Q)$ over $f \in \mathcal{P}$.
Choose $\hat{h}$ to minimize $\hat{R}_{S_Q}(h)$ among $h \in \mathcal{H}$
subject to $\hat{\mathcal{E}}_{S_P, \hat{f}}(h) \leq c\sqrt{\hat{P}_{S_P, \hat{f}^2}(h \neq \hat{h}_{S_P, \hat{f}}) A_{n_P}''} + c\|\hat{f}\|_\infty A_{n_P}''$.

---

As we establish in the proof, $\hat{f}$ is effectively being chosen to minimize an upper bound on the excess $Q$-risk of the resulting classifier $\hat{h}$. Toward analyzing the performance of this procedure, note that each $f$ induces a marginal transfer exponent: that is, values $C_{\gamma, f}, \gamma_f$ such that $\forall h \in \mathcal{H}$, $C_{\gamma, f} P_{f^2}(h \neq h_{P_f}^*) \geq Q^{\gamma_f}(h \neq h_{P_f}^*)$. Similarly, each $f$ induces a Bernstein Class Condition: there exist values $c_f > 0$, $\beta_f \in [0, 1]$ such that $P_{f^2}(h \neq h_{P_f}^*) \leq c_f \mathcal{E}_{P_f}^{\beta_f}(h)$.

The following theorem reveals that Algorithm 3 is able to perform nearly as well as applying the transfer technique from Theorem 3 directly under the measure in the family $\mathcal{P}$ that would provide the best bound. The only losses compared to doing so are a dependence on $d_p$ and the supremum of the density (which accounts for how different that measure is from $P$). The proof is in Appendix E.

**Theorem 5.** *Suppose $\beta_Q > 0$ and that (NC) and (RCS) hold for all $P_f$, $f \in \mathcal{P}$. There exist constants $C_f$ depending on $\|f\|_\infty$, $C_{\gamma, f}$, $\gamma_f$, $c_f$, $\beta_f$, and a constant $C$ depending on $c_q$, $\beta_Q$ such that, for a sufficiently large $|U_Q|$, w.p. at least $1 - \delta$, the classifier $\hat{h}$ chosen by Algorithm 3 satisfies*

$$\mathcal{E}_Q(\hat{h}) \leq \inf_{f \in \mathcal{P}} C \min\left\{C_f\left(A_{n_P}''\right)^{\frac{\beta_f}{(2 - \beta_f)\gamma_f}}, A_{n_Q}^{\frac{1}{2 - \beta_Q}}\right\}$$

$$= \tilde{O}\left(\inf_{f \in \mathcal{P}} \min\left\{C_f\left(\frac{d_{\mathcal{H}} + d_p}{n_P}\right)^{\frac{\beta_f}{(2 - \beta_f)\gamma_f}}, \left(\frac{d_{\mathcal{H}}}{n_Q}\right)^{\frac{1}{2 - \beta_Q}}\right\}\right).$$

The utility of this theorem will of course depend largely on the family $\mathcal{P}$ of densities. This class should contain a distribution with small $\gamma_f$ marginal transfer exponent, while also small $\|f\|_\infty$ (which is captured by the $C_f$ constant in the bound), and favorable noise conditions (i.e., large $\beta_f$).

## A.2 Choice of Transfer from Multiple Sources

It is worth noting that all of the above analysis also applies to the case that, instead of a family of densities with respect to a single $P$, the set $\mathcal{P}$ is a set of probability measures $P_i$, each with its own separate iid data set $S_i$ of some size $n_i$. Lemma 1 can then be applied to all of these data sets, if we simply replace $\delta$ by $\delta/|\mathcal{P}|$ to accommodate a union bound; call the corresponding quantity $A_n'''$. Then, similarly to the above, we can use the following procedure.

---

Algorithm 4:
Choose $\hat{i}$ to minimize $\hat{\delta}(S_i, U_Q)$ over $P_i \in \mathcal{P}$.
Choose $\hat{h}$ to minimize $\hat{R}_{S_Q}(h)$ among $h \in \mathcal{H}$
subject to $\hat{\mathcal{E}}_{S_{\hat{i}}}(h) \leq c\sqrt{\hat{P}_{S_{\hat{i}}}(h \neq \hat{h}_{S_{\hat{i}}}) A_{n_{\hat{i}}}'''} + c A_{n_{\hat{i}}}'''$.

---

To state a formal guarantee, let us suppose the conditions above hold for each of these distributions with respective values of $C_{\gamma,i}$, $\gamma_i$, $c_i$, $\beta_i$. We have the following theorem. Its proof is essentially identical to the proof of Theorem 5 (effectively just substituting notation), and is therefore omitted.

**Theorem 6.** *Suppose $\beta_Q > 0$ and that (NC) and (RCS) hold for all $P_i \in \mathcal{P}$. There exist constants $C_i$ depending on $C_{\gamma,i}$, $\gamma_i$, $c_i$, $\beta_i$, and a constant $C$ depending on $c_q$, $\beta_Q$ such that, for a sufficiently large $|U_Q|$, with probability at least $1 - \delta$, the classifier $\hat{h}$ chosen by Algorithm 4 satisfies*

$$\mathcal{E}_Q(\hat{h}) \leq \tilde{O}\left( \inf_{P_i \in \mathcal{P}} \min\left\{ C_i \left( \frac{d_\mathcal{H} + \log(|\mathcal{P}|)}{n_i} \right)^{\frac{\beta_i}{(2-\beta_i)\gamma_i}}, \left( \frac{d_\mathcal{H}}{n_Q} \right)^{\frac{1}{2-\beta_Q}} \right\} \right).$$

# B Lower-Bounds Proofs

Our lower-bounds rely on the following extensions of Fano inequality.

**Proposition 4** (Thm 2.5 of [33]). *Let $\{\Pi_h\}_{h \in \mathcal{H}}$ be a family of distributions indexed over a subset $\mathcal{H}$ of a semi-metric $(\mathcal{F}, \mathrm{dist})$. Suppose $\exists\, h_0, \ldots, h_M \in \mathcal{H}$, where $M \geq 2$, such that:*

 (i) $\mathrm{dist}(h_i, h_j) \geq 2s > 0, \quad \forall 0 \leq i < j \leq M,$
 (ii) $\Pi_{h_i} \ll \Pi_{h_0} \quad \forall i \in [M]$, *and the average KL-divergence to $\Pi_{h_0}$ satisfies*

$$\frac{1}{M} \sum_{i=1}^M \mathcal{D}_{kl}\left(\Pi_{h_i} | \Pi_{h_0}\right) \leq \alpha \log M, \text{ where } 0 < \alpha < 1/8.$$

*Let $Z \sim \Pi_h$, and let $\hat{h} : Z \mapsto \mathcal{F}$ denote any* improper *learner of $h \in \mathcal{H}$. We have for any $\hat{h}$:*

$$\sup_{h \in \mathcal{H}} \Pi_h\left(\mathrm{dist}\left(\hat{h}(Z), h\right) \geq s\right) \geq \frac{\sqrt{M}}{1 + \sqrt{M}}\left(1 - 2\alpha - \sqrt{\frac{2\alpha}{\log(M)}}\right) \geq \frac{3 - 2\sqrt{2}}{8}.$$

The following proposition would be needed to construct packings (of spaces of distributions) of the appropriate size.

**Proposition 5** (Varshamov-Gilbert bound). *Let $d \geq 8$. Then there exists a subset $\{\sigma_0, \ldots, \sigma_M\}$ of $\{-1, 1\}^d$ such that $\sigma_0 = (1, \ldots, 1)$,*

$$\mathrm{dist}(\sigma_i, \sigma_j) \geq \frac{d}{8}, \quad \forall\, 0 \leq i < j \leq M, \quad \text{and} \quad M \geq 2^{d/8},$$

*where $\mathrm{dist}(\sigma, \sigma') \doteq \mathrm{card}(\{i \in [m] : \sigma(i) \neq \sigma'(i)\})$ is the Hamming distance.*

Results similar to the following lemma are known.

**Lemma 2** (A basic KL upper-bound). *For any* $0 < p, q < 1$, *we let* $\mathcal{D}_{kl}(p|q)$ *denote* $\mathcal{D}_{kl}(Ber(p)|Ber(q))$. *Now let* $0 < \epsilon < 1/2$ *and let* $z \in \{-1, 1\}$. *We have*

$$\mathcal{D}_{kl}(1/2 + (z/2) \cdot \epsilon \,|\, 1/2 - (z/2) \cdot \epsilon) \leq c_0 \cdot \epsilon^2, \text{ for some } c_0 \text{ independent of } \epsilon.$$

*Proof.* Write $\frac{p}{q} \doteq \frac{1/2 + (z/2)\epsilon}{1/2 - (z/2)\epsilon} = 1 + \frac{2z\epsilon}{1 - z\epsilon}$, and use the fact that

$$\mathcal{D}_{kl}(p|q) \leq \chi^2(p|q) = q\left(1 - \frac{p}{q}\right)^2 + (1-q)\left(1 - \frac{1-p}{1-q}\right)^2 = q\left(\frac{2z\epsilon}{1-z\epsilon}\right)^2 + (1-q)\left(\frac{-2z\epsilon}{1+z\epsilon}\right)^2.$$

$\square$

*Proof of Theorem 1.* Let $d = d_{\mathcal{H}} - 1$. Pick $x_0, x_1, x_2, \ldots, x_d$ a shatterable subset of $\mathcal{X}$ under $\mathcal{H}$. These will form the support of marginals $P_X, Q_X$. Furthermore, let $\tilde{\mathcal{H}}$ denote the *projection* of $\mathcal{H}$ onto $\{x_i\}_{i=0}^d$ (i.e., the quotient space of equivalences $h \equiv h'$ on $\{x_i\}$), with the additional constraint that all $h \in \tilde{\mathcal{H}}$ classify $x_0$ as 1. We can now restrict attention to $\tilde{\mathcal{H}}$ as the *effective* hypothesis class.

Let $\sigma \in \{-1, 1\}^d$. We will construct a family of distribution pairs $(P_\sigma, Q_\sigma)$ indexed by $\sigma$ to which we then apply Proposition 4 above. For any $P_\sigma, Q_\sigma$, we let $\eta_{P,\sigma}, \eta_{Q,\sigma}$ denote the corresponding regression functions (i.e., $\mathbb{E}_{P_\sigma}[Y|x]$, and $\mathbb{E}_{Q_\sigma}[Y|x]$). To proceed, fix $\epsilon = c_1 \cdot \epsilon(n_P, n_Q) \leq 1/2$, for a constant $c_1 < 1$ to be determined, where $\epsilon(n_P, n_Q)$ is as defined in the theorem's statement.

- *Distribution $Q_\sigma$.* We have that $Q_\sigma = Q_X \times Q_{Y|X}^\sigma$, where $Q_X(x_0) = 1 - \epsilon^{\beta_Q}$, while $Q_X(x_i) = \frac{1}{d}\epsilon^{\beta_Q}$, $i \geq 1$. Now, the conditional $Q_{Y|X}^\sigma$ is fully determined by $\eta_{Q,\sigma_i}(x_0) = 1$, and $\eta_{Q,\sigma}(x_i) = 1/2 + (\sigma_i/2) \cdot \epsilon^{1-\beta_Q}$, $i \geq 1$.

- *Distribution $P_\sigma$.* We have that $P_\sigma = P_X \times P_{Y|X}^\sigma$, $P_X(x_0) = 1 - \epsilon^{\rho\beta_P}$, while $P_X(x_i) = \frac{1}{d}\epsilon^{\rho\beta_P}$, $i \geq 1$. Now, the conditional $P_{Y|X}^\sigma$ is fully determined by $\eta_{P,\sigma}(x_0) = 1$, and $\eta_{P,\sigma}(x_i) = 1/2 + (\sigma_i/2) \cdot \epsilon^{\rho(1-\beta_P)}$, $i \geq 1$.

- *Verifying that $(P_\sigma, Q_\sigma) \in \mathcal{F}_{(NC)}(\rho, \beta_P, \beta_Q, 1)$.* For any $\sigma \in \{-1, 1\}^d$, let $h_\sigma \in \tilde{\mathcal{H}}$ denote the corresponding Bayes classifier (remark that the Bayes is the same for both $P_\sigma$ and $Q_\sigma$). Now, pick any other $h_{\sigma'} \in \tilde{\mathcal{H}}$, and let $\text{dist}(\sigma, \sigma')$ denote the Hamming distance between $\sigma, \sigma'$ (as in Proposition 5). We then have that

$$\mathcal{E}_{Q_\sigma}(h_{\sigma'}) = \text{dist}(\sigma, \sigma') \cdot \frac{1}{d}\epsilon^{\beta_Q} \cdot \epsilon^{1-\beta_Q} = \frac{\text{dist}(\sigma, \sigma')}{d} \cdot \epsilon,$$

$$\text{while } Q_X(h_{\sigma'} \neq h_\sigma) = \frac{\text{dist}(\sigma, \sigma')}{d} \cdot \epsilon^{\beta_Q},$$

$$\text{and similarly, } \mathcal{E}_{P_\sigma}(h_{\sigma'}) = \frac{\text{dist}(\sigma, \sigma')}{d} \cdot \epsilon^\rho, \text{ while } P_X(h_{\sigma'} \neq h_\sigma) = \frac{\text{dist}(\sigma, \sigma')}{d} \cdot \epsilon^{\rho\beta_P}.$$

The condition is also easily verified for classifiers not labeling $x_0$ as 1. Since $(\text{dist}(\sigma, \sigma')/d) \leq 1$, it follows that (1) holds with exponents $\beta_P$ and $\beta_Q$ for any $P_\sigma$ and $Q_\sigma$ respectively (with $C_{P_\sigma} = 1$, $C_{Q_\sigma} = 1$), and that any $P_\sigma$ admits a transfer-exponent $\rho$ w.r.t. $Q_\sigma$, with $C_\rho = 1$.

- *Reduction to a packing.* Now apply Proposition 5 to identify a subset $\Sigma$ of $\{-1, 1\}^d$, where $|\Sigma| = M \geq 2^{d/8}$, and $\forall \sigma, \sigma' \in \Sigma$, we have $\text{dist}(\sigma, \sigma') \geq d/8$. It should be clear then that for any $\sigma, \sigma' \in \Sigma$,

$$\mathcal{E}_{Q_\sigma}(h_{\sigma'}) \geq \frac{d}{8} \cdot \frac{1}{d}\epsilon^{\beta_Q} \cdot \epsilon^{1-\beta_Q} = \epsilon/8.$$

Furthermore, by construction, any classifier $\hat{h} : \{x_i\} \mapsto \{0, 1\}$ can be reduced to a decision on $\sigma$, and we henceforth view $\text{dist}(\sigma, \sigma')$ as the semi-metric referenced in Proposition 4, with effective indexing set $\Sigma$.

- *KL bounds in terms of $n_P$ and $n_Q$.* Define $\Pi_\sigma = P_\sigma^{n_P} \times Q_\sigma^{n_Q}$. We can now verify that all $\Pi_\sigma, \Pi_{\sigma'}$ are close in KL-divergence. First notice that, for any $\sigma, \sigma' \in \Sigma$ (in fact in $\{-1,1\}^d$)

$$
\begin{aligned}
\mathcal{D}_{\mathrm{kl}}\left(\Pi_\sigma | \Pi_{\sigma'}\right) &= n_P \cdot \mathcal{D}_{\mathrm{kl}}\left(P_\sigma | P_{\sigma'}\right) + n_Q \cdot \mathcal{D}_{\mathrm{kl}}\left(Q_\sigma | Q_{\sigma'}\right) \\
&= n_P \cdot \underset{P_X}{\mathbb{E}}\, \mathcal{D}_{\mathrm{kl}}\left(P_{Y|X}^\sigma | P_{Y|X}^{\sigma'}\right) + n_Q \cdot \underset{Q_X}{\mathbb{E}}\, \mathcal{D}_{\mathrm{kl}}\left(Q_{Y|X}^\sigma | Q_{Y|X}^{\sigma'}\right) \\
&= n_P \cdot \sum_{i=1}^d \frac{\epsilon^{\rho\beta_P}}{d} \mathcal{D}_{\mathrm{kl}}\left(P_{Y|x_i}^\sigma | P_{Y|x_i}^{\sigma'}\right) + n_Q \cdot \sum_{i=1}^d \frac{\epsilon^{\beta_Q}}{d} \mathcal{D}_{\mathrm{kl}}\left(Q_{Y|x_i}^\sigma | Q_{Y|x_i}^{\sigma'}\right) \\
&\leq c_0 \left(n_P \cdot \epsilon^{\rho(2-\beta_P)} + n_Q \cdot \epsilon^{(2-\beta_Q)}\right) \qquad (9) \\
&\leq c_0 d (c_1^{\rho(2-\beta_P)} + c_1^{2-\beta_Q}) \leq 2 c_0 c_1 d. \qquad (10)
\end{aligned}
$$

where, for inequality (9), we used Lemma 2 to upper-bound the divergence terms. It follows that, for $c_1$ sufficiently small so that $2 c_0 c_1 \leq 1/16$, we get that (10) is upper bounded by $(1/8) \log M$. Now apply Proposition 4 and conclude. $\qquad\square$

We need the following lemma for the next result.

**Lemma 3.** *Let $\epsilon_1, \epsilon_2, \alpha, \alpha_1, \alpha_2 \geq 0$, and $\alpha_1 + \alpha_2 \leq 1$. We then have that*

$$
\begin{aligned}
\text{For } \alpha \geq 1, \quad &\alpha_1 \epsilon_1^\alpha + \alpha_2 \epsilon_2^\alpha \geq (\alpha_1 \epsilon_1 + \alpha_2 \epsilon_2)^\alpha, \text{ and} \\
\text{for } \alpha \leq 1, \quad &\alpha_1 \epsilon_1^\alpha + \alpha_2 \epsilon_2^\alpha \leq (\alpha_1 \epsilon_1 + \alpha_2 \epsilon_2)^\alpha.
\end{aligned}
$$

*Proof.* W.l.o.g., let $\alpha_1 + \alpha_2 > 0$, and normalize the l.h.s. of each of the above inequalities by $(\alpha_1 + \alpha_2)^{-1} \geq 1$. The results follows by Jensen's inequality and the convexity of $z \mapsto z^\alpha$ for $\alpha \geq 1$, and concavity of $z \mapsto z^\alpha$ for $\alpha \leq 1$. $\qquad\square$

We can now show Theorem 2.

*Proof of Theorem 2.* We proceed similarly (as far as high-level arguments) as for the proof of Theorem 1, but with a different construction where distributions now all satisfy $\gamma = \rho \cdot \beta_P$, and are broken into two subfamilies (corresponding to the rates $\epsilon_1$ and $\epsilon_2$), and the final result holds by considering the intersection of these subfamilies. For simplicity, in what follows, assume $d$ is even, otherwise, the arguments hold by just replacing $d$ by $d-1$. First, define $x_0, x_1, x_2, \ldots, x_d, \tilde{\mathcal{H}}$ as in that proof.

Let $\sigma \in \{-1,1\}^d$. Next we construct distribution pairs $P_\sigma, Q_\sigma$ indexed by $\sigma$, with corresponding regression functions $\eta_{P,\sigma}, \eta_{Q,\sigma}$. Fix $\epsilon_1 = c_1 \cdot \epsilon_1(n_P, n_Q) \leq 1/2$, and $\epsilon_2 = c_2 \cdot \epsilon_2(n_P, n_Q) \leq 1/2$, for some $c_1, c_2 < 1$ to be determined.

The construction is now broken up over $I_1 \doteq \left\{1, \ldots, \frac{d}{2}\right\}$, and $I_2 \doteq \left\{\frac{d}{2}+1, \ldots, d\right\}$. Fix a constant $\frac{1}{2} \leq \tau < 1$; this ensures that $\epsilon_2/\tau \leq 1$. We will later impose further conditions on $\tau$.

- *Distribution $Q_\sigma$.* We let $Q_\sigma = Q_X \times Q_{Y|X}^\sigma$, where $Q_X(x_0) = 1 - \frac{1}{2}\left(\epsilon_1^{\beta_Q} + (\epsilon_2/\tau)\right)$, while $Q_X(x_i) = \frac{1}{d}\epsilon_1^{\beta_Q}$ for $i \in I_1$, and $Q_X(x_i) = \frac{1}{d}(\epsilon_2/\tau)$ for $i \in I_2$. Now, the conditional $Q_{Y|X}^\sigma$ is fully determined by $\eta_{Q,\sigma}(x_0) = 1$, and $\eta_{Q,\sigma}(x_i) = 1/2 + (\sigma_i/2) \cdot \epsilon_1^{1-\beta_Q}$ for $i \in I_1$, and $\eta_{Q,\sigma}(x_i) = 1/2 + (\sigma_i/2) \cdot \tau$ for $i \in I_2$.

- *Distribution $P_\sigma$.* We let $P_\sigma = P_X \times P_{Y|X}^\sigma$, where $P_X(x_0) = 1 - \frac{1}{2}\left(\epsilon_1^{\gamma\beta_Q} + \epsilon_2^\gamma\right)$, while $P_X(x_i) = \frac{1}{d}\epsilon_1^{\gamma\beta_Q}$ for $i \in I_1$, and $P_X(x_i) = \frac{1}{d}\epsilon_2^\gamma$ for $i \in I_2$. Now, the conditional $P_{Y|X}^\sigma$ is fully determined by $\eta_{P,\sigma}(x_0) = 1$, and $\eta_{P,\sigma}(x_i) = 1/2 + (\sigma_i/2) \cdot \epsilon_1^{(1-\beta_P)\rho\beta_Q}$ for $i \in I_1$, and $\eta_{P,\sigma}(x_i) = 1/2 + (\sigma_i/2) \cdot \epsilon_2^{(1-\beta_P)\rho}$ for $i \in I_2$.

- *Verifying that $(P_\sigma, Q_\sigma) \in \mathcal{F}_{(NC)}(\rho, \beta_P, \beta_Q, 2)$.* For any $\sigma \in \{-1,1\}^d$, define $h_\sigma \in \tilde{\mathcal{H}}$ as in the proof of Theorem 1. Now, pick any other $h_{\sigma'} \in \tilde{\mathcal{H}}$, and let $\mathrm{dist}_I(\sigma, \sigma')$ denote the Hamming distance

between $\sigma, \sigma'$, restricted to indices in $I$ (that is the Hamming distance between subvectors $\sigma_I$ and $\sigma'_I$). We then have that

$$\mathcal{E}_{Q_\sigma}(h_{\sigma'}) = \text{dist}_{I_1}(\sigma, \sigma') \cdot \frac{1}{d} \epsilon^{\beta_Q} \cdot \epsilon_1^{1-\beta_Q} + \text{dist}_{I_2}(\sigma, \sigma') \cdot \frac{1}{d}(\epsilon_2/\tau)\tau$$

$$= \frac{\text{dist}_{I_1}(\sigma, \sigma')}{d}\epsilon_1 + \frac{\text{dist}_{I_2}(\sigma, \sigma')}{d}\epsilon_2,$$

$$\text{while } Q_X(h_{\sigma'} \neq h_\sigma) = \frac{\text{dist}_{I_1}(\sigma, \sigma')}{d}\epsilon_1^{\beta_Q} + \frac{\text{dist}_{I_2}(\sigma, \sigma')}{d}(\epsilon_2/\tau).$$

$$\text{Similarly, } \mathcal{E}_{P_\sigma}(h_{\sigma'}) = \text{dist}_{I_1}(\sigma, \sigma') \cdot \frac{1}{d}\epsilon_1^{\gamma\beta_Q} \cdot \epsilon_1^{(1-\beta_P)\rho\beta_Q} + \text{dist}_{I_2}(\sigma, \sigma') \cdot \frac{1}{d}\epsilon_2^\gamma \cdot \epsilon_2^{(1-\beta_P)\rho},$$

$$= \frac{\text{dist}_{I_1}(\sigma, \sigma')}{d}\epsilon_1^{\rho\beta_Q} + \frac{\text{dist}_{I_2}(\sigma, \sigma')}{d}\epsilon_2^\rho,$$

$$\text{while } P_X(h_{\sigma'} \neq h_\sigma) = \frac{\text{dist}_{I_1}(\sigma, \sigma')}{d}\epsilon_1^{\gamma\beta_Q} + \frac{\text{dist}_{I_2}(\sigma, \sigma')}{d}\epsilon_2^\gamma.$$

The condition is also easily verified for classifiers not labeling $x_0$ as 1. We apply Lemma 3 repeatedly in what follows. First, by the above, we have that

$$Q_X(h_{\sigma'} \neq h_\sigma) \leq \frac{\text{dist}_{I_1}(\sigma, \sigma')}{d}\epsilon_1^{\beta_Q} + 2\frac{\text{dist}_{I_2}(\sigma, \sigma')}{d}\epsilon_2^{\beta_Q} \leq 2\mathcal{E}_{Q_\sigma}^{\beta_Q}(h_{\sigma'}).$$

On the other hand,

$$P_X(h_{\sigma'} \neq h_\sigma) = \frac{\text{dist}_{I_1}(\sigma, \sigma')}{d}\left(\epsilon_1^{\rho\beta_Q}\right)^{\beta_P} + \frac{\text{dist}_{I_2}(\sigma, \sigma')}{d}(\epsilon_2^\rho)^{\beta_P} \leq \mathcal{E}_{P_\sigma}^{\beta_P}(h_{\sigma'}),$$

Finally we have that

$$\mathcal{E}_{P_\sigma}(h_{\sigma'}) \geq \frac{\text{dist}_{I_1}(\sigma, \sigma')}{d}\epsilon_1^\rho + \frac{\text{dist}_{I_2}(\sigma, \sigma')}{d}\epsilon_2^\rho \geq \mathcal{E}_{Q_\sigma}^\rho(h_{\sigma'}).$$

- *Verifying that $\gamma$ is a marginal-transfer-exponent $P_X$ to $Q_X$.* Using the above derivations, the condition that $\gamma \geq 1$, and further imposing the condition that $\tau \geq (1/2)^{1/\gamma}$, we have

$$P_X(h_{\sigma'} \neq h_\sigma) \geq \frac{\text{dist}_{I_1}(\sigma, \sigma')}{d}\left(\epsilon_1^{\beta_Q}\right)^\gamma + \frac{1}{2}\frac{\text{dist}_{I_2}(\sigma, \sigma')}{d}(\epsilon_2/\tau)^\gamma \geq \frac{1}{2}Q_X^\gamma(h_{\sigma'} \neq h_\sigma).$$

where we again used Lemma 3.

- *Reduction to sub-Packings.* Now, in a slight deviation from the proof of Theorem 1, we define two separate packings (in Hamming distance), indexed by some $\varsigma$ as follows. Fix any $\varsigma \in \{-1, 1\}^{d/2}$, and applying Proposition 2, let $\Sigma_1(\varsigma) \subset \left\{\sigma \in \{-1, 1\}^d : \sigma_{I_2} = \varsigma\right\}$, and $\Sigma_2(\varsigma) \subset \left\{\sigma \in \{-1, 1\}^d : \sigma_{I_1} = \varsigma\right\}$ denote $m$-packings of $\{-1, 1\}^{d/2}$, $m \geq d/16$, of size $M + 1$, $M \geq 2^{d/16}$.

Clearly, for any $\sigma, \sigma' \in \Sigma_1(\varsigma)$ we have $\mathcal{E}_{Q_\sigma}(h_{\sigma'}) \geq \epsilon_1/16$, while for any $\sigma, \sigma' \in \Sigma_2(\varsigma)$ we have $\mathcal{E}_{Q_\sigma}(h_{\sigma'}) \geq \epsilon_2/16$.

- *KL Bounds in terms of $n_P$ and $n_Q$.* Again, define $\Pi_\sigma = P_\sigma^{n_P} \times Q_\sigma^{n_Q}$. First, for any $\varsigma$ fixed, let $\sigma, \sigma' \in \Sigma_1(\varsigma)$. As in the proof of Theorem 1, we apply Lemma 2 to get that

$$\mathcal{D}_{\text{kl}}(\Pi_\sigma | \Pi_{\sigma'}) = n_P \cdot \underset{P_X}{\mathbb{E}} \, \mathcal{D}_{\text{kl}}\left(P_{Y|X}^\sigma | P_{Y|X}^{\sigma'}\right) + n_Q \cdot \underset{Q_X}{\mathbb{E}} \, \mathcal{D}_{\text{kl}}\left(Q_{Y|X}^\sigma | Q_{Y|X}^{\sigma'}\right)$$

$$= n_P \cdot \sum_{i \in I_1} \frac{\epsilon_1^{\gamma\beta_Q}}{d}\mathcal{D}_{\text{kl}}\left(P_{Y|x_i}^\sigma | P_{Y|x_i}^{\sigma'}\right) + n_Q \cdot \sum_{i \in I_1} \frac{\epsilon_1^{\beta_Q}}{d}\mathcal{D}_{\text{kl}}\left(Q_{Y|x_i}^\sigma | Q_{Y|x_i}^{\sigma'}\right)$$

$$\leq n_P \cdot c_0 \frac{1}{2}\epsilon_1^{(2-\beta_P)\rho\beta_Q} + n_Q \cdot c_0 \frac{1}{2}\epsilon_1^{(2-\beta_Q)}$$

$$\leq c_0 \frac{d}{2}(c_1^{(2-\beta_P)\rho\beta_Q} + c_1^{2-\beta_Q}) \leq c_0 c_1 d.$$

Similarly, for any $\varsigma$ fixed, let $\sigma, \sigma' \in \Sigma_2(\varsigma)$; expanding over $I_2$, we have:

$$\mathcal{D}_{\mathrm{kl}}\left(\Pi_\sigma | \Pi_{\sigma'}\right) \le n_P \cdot c_0 \frac{1}{2}\epsilon_2^{(2-\beta_P)\rho} + n_Q \cdot c_0 \frac{1}{2}\epsilon_2 \cdot \tau \le c_0 c_1 d.$$

It follows that, for $c_1$ sufficiently small so that $c_0 c_1 \le 1/16$, we can apply Proposition 4 twice, to get that *for all* $\varsigma$, there exist $\sigma_{I_1}$ and $\sigma_{I_2}$, such that for some constant $c$, we have

$$\mathbb{E}_{\Pi_\sigma}\left(\mathcal{E}_{Q_\sigma}(\hat{h})\right) \ge c \cdot \epsilon_1, \text{ where } \sigma = [\sigma_{I_1}, \varsigma], \text{ and } \mathbb{E}_{\Pi_\sigma}\left(\mathcal{E}_{Q_\sigma}(\hat{h})\right) \ge c \cdot \epsilon_2, \text{ where } \sigma = [\varsigma, \sigma_{I_2}].$$

It follows that $c \cdot \max\{\epsilon_1, \epsilon_2\}$ is a lower-bound for either $\sigma = [\sigma_{I_1}, \varsigma]$ or $\sigma = [\varsigma, \sigma_{I_2}]$. $\qquad\square$

## C   Upper Bounds Proofs

*Proof of Proposition 2.* To reduce redundancy, we refer to arguments presented in the proof of Theorem 3, rather than repeating them here. As in the proof of Theorem 3, we let $C$ serve as a generic constant (possibly depending on $\rho', C_{\rho'}, \beta_P, c_{\beta_P}, \beta_Q, c_{\beta_Q}$) which may be different in different appearances. Define a set

$$\mathcal{G} = \left\{ h \in \mathcal{H} : \hat{R}_{S_Q}(h) - \hat{R}_{S_Q}(\hat{h}_{S_Q}) \le c\sqrt{\hat{P}_{S_Q}(h \ne \hat{h}_{S_Q})A_{n_Q}} + cA_{n_Q} \right\}.$$

We can rephrase the definition of $\hat{h}$ as saying $\hat{h} = \hat{h}_{S_P}$ when $\hat{h}_{S_P} \in \mathcal{G}$, and otherwise $\hat{h} = \hat{h}_{S_Q}$.

We suppose the event from Lemma 1 holds for both $S_Q$ and $S_P$; by the union bound, this happens with probability at least $1 - \delta$. In particular, as in (8) from the proof of Theorem 3, we have

$$\mathcal{E}_P(\hat{h}_{S_P}) \le CA_{n_P}^{\frac{1}{2-\beta_P}}.$$

Together with the definition of $\rho'$, this implies

$$\mathcal{E}_Q(\hat{h}_{S_P}, h_P^*) \le CA_{n_P}^{\frac{1}{(2-\beta_P)\rho'}},$$

which means

$$\mathcal{E}_Q(\hat{h}_{S_P}) \le \mathcal{E}_Q(h_P^*) + \mathcal{E}_Q(\hat{h}_{S_P}, h_P^*) \le \mathcal{E}_Q(h_P^*) + CA_{n_P}^{\frac{1}{(2-\beta_P)\rho'}}. \tag{11}$$

Now, if $R_Q(\hat{h}_{S_P}) \le R_Q(\hat{h}_{S_Q})$, then (due to the event from Lemma 1) we have $\hat{h}_{S_P} \in \mathcal{G}$, so that $\hat{h} = \hat{h}_{S_P}$, and thus the rightmost expression in (11) bounds $\mathcal{E}_Q(\hat{h})$. On the other hand, if $R_Q(\hat{h}_{S_P}) > R_Q(\hat{h}_{S_Q})$, then regardless of whether $\hat{h} = \hat{h}_{S_P}$ or $\hat{h} = \hat{h}_{S_Q}$, we have $\mathcal{E}_Q(\hat{h}) \le \mathcal{E}_Q(\hat{h}_{S_P})$, so that again the rightmost expression in (11) bounds $\mathcal{E}_Q(\hat{h})$. Thus, in either case,

$$\mathcal{E}_Q(\hat{h}) \le \mathcal{E}_Q(h_P^*) + CA_{n_P}^{\frac{1}{(2-\beta_P)\rho'}}.$$

Furthermore, as in the proof of Theorem 3, every $h \in \mathcal{G}$ satisfies $\mathcal{E}_Q(h) \le CA_{n_Q}^{\frac{1}{2-\beta_Q}}$. Since the algorithm only picks $\hat{h} = \hat{h}_{S_P}$ if $\hat{h}_{S_P} \in \mathcal{G}$, and otherwise picks $\hat{h} = \hat{h}_{S_Q}$, which is clearly in $\mathcal{G}$, we may note that we always have $\hat{h} \in \mathcal{G}$. We therefore conclude that

$$\mathcal{E}_Q(\hat{h}) \le CA_{n_Q}^{\frac{1}{2-\beta_Q}},$$

which completes the proof. $\qquad\square$

## D   Proofs for Adaptive Sampling Costs

*Proof of Theorem 4.* First note that since $\sum_n \frac{1}{2n^2} < 1$, by the union bound and Lemma 1, with probability at least $1 - \delta$, for every $h, h' \in \mathcal{H}$, every set $S_P$ in the algorithm has

$$R_P(h) - R_P(h') \le \hat{R}_{S_P}(h) - \hat{R}_{S_P}(h') + c\sqrt{\min\{P(h \ne h'), \hat{P}_{S_P}(h \ne h')\}A'_{|S_P|}} + cA'_{|S_P|}$$

and

$$\hat{P}_{S_P}(h \neq h') \leq 2P(h \neq h') + cA'_{|S_P|}$$

every set $S_Q$ in the algorithm has

$$R_Q(h) - R_Q(h') \leq \hat{R}_{S_Q}(h) - \hat{R}_{S_Q}(h') + c\sqrt{\min\{Q(h \neq h'), \hat{P}_{S_Q}(h \neq h')\}A'_{|S_Q|}} + cA'_{|S_Q|}$$

and

$$\hat{P}_{S_Q}(h \neq h') \leq 2Q(h \neq h') + cA'_{|S_Q|},$$

and we also have for the set $U_Q$ that

$$\frac{1}{2}Q(h \neq h') - cA_{|U_Q|} \leq \hat{P}_{U_Q}(h \neq h') \leq 2Q(h \neq h') + cA_{|U_Q|},$$

which by our choice of the size of $U_Q$ implies

$$\frac{1}{2}Q(h \neq h') - \frac{\epsilon}{8} \leq \hat{P}_{U_Q}(h \neq h') \leq 2Q(h \neq h') + \frac{\epsilon}{8}.$$

For the remainder of this proof, we suppose these inequalities hold.

In particular, these imply

$$R_Q(\hat{h}_{S_Q}) - R_Q(h^*) \leq c\sqrt{\hat{P}_{S_Q}(\hat{h}_{S_Q} \neq h^*)A'_{|S_Q|}} + cA'_{|S_Q|}.$$

Furthermore,

$$\hat{R}_{S_Q}(h^*) - \hat{R}_{S_Q}(\hat{h}_{S_Q}) \leq c\sqrt{\hat{P}_{S_Q}(h^* \neq \hat{h}_{S_Q})A'_{|S_Q|}} + cA'_{|S_Q|},$$

so that $h = h^*$ is included in the supremum in the definition of $\hat{\delta}(S_Q, S_Q)$. Together these imply

$$\mathcal{E}_Q(\hat{h}_{S_Q}) \leq R_Q(\hat{h}_{S_Q}) - R_Q(h^*) \leq c\sqrt{\hat{\delta}(S_Q, S_Q)A_{|S_Q|}} + cA_{|S_Q|}.$$

Thus, if the algorithm returns $\hat{h}_{S_Q}$ in Step 6, then $\mathcal{E}_Q(\hat{h}_{S_Q}) \leq \epsilon$.

Also by the above inequalities, we have

$$\hat{R}_{S_P}(h^*) - \hat{R}_{S_P}(\hat{h}_{S_P}) \leq c\sqrt{\hat{P}_{S_P}(h^* \neq \hat{h}_{S_Q})A'_{|S_P|}} + cA'_{|S_P|},$$

so that $h^*$ is included in the supremum in the definition of $\hat{\delta}(S_P, U_Q)$. Thus,

$$\mathcal{E}_Q(\hat{h}_{S_P}) \leq Q(\hat{h}_{S_P} \neq h^*) \leq 2\hat{P}_{U_Q}(\hat{h}_{S_P} \neq h^*) + \frac{\epsilon}{2} \leq 2\hat{\delta}(S_P, U_Q) + \frac{\epsilon}{2},$$

and hence if the algorithm returns $\hat{h}_{S_P}$ in Step 7 we have $\mathcal{E}_Q(\hat{h}_{S_P}) \leq \epsilon$ as well. Furthermore, the algorithm will definitely return at some point, since the bound in Step 6 approaches $0$ as the sample size grows. Altogether, this establishes that, on the above event, the $\hat{h}$ returned by the algorithm satisfies $\mathcal{E}_Q(\hat{h}) \leq \epsilon$, as claimed.

It remains to show that the cost satisfies the stated bound. For this, first note that since the costs incurred by the algorithm grow as a function that is upper and lower bounded by a geometric series, it suffices to argue that, for an appropriate choice of the constant $c'$, the algorithm would halt if ever it reached a set $S_P$ of size at least $n_P^*$ or a set $S_Q$ of size at least $n_Q^*$ (which ever were to happen first); the result would then follow by choosing the actual constant $c'$ in the theorem slightly larger than this, to account for the algorithm slighly "overshooting" this target (by at most a numerical constant factor).

First suppose it reaches $S_Q$ of size at least $n_Q^*$. Now, as in the proof of Theorem 3, on the above event, every $h \in \mathcal{H}$ included in the supremum in the definition of $\hat{\delta}(S_Q, S_Q)$ has

$$\mathcal{E}_Q(h) \leq C\left(A'_{|S_Q|}\right)^{\frac{1}{2-\beta_Q}},$$

which further implies

$$Q(h \neq h^*) \leq C \left( A'_{|S_Q|} \right)^{\frac{\beta_Q}{2 - \beta_Q}},$$

so that (by the triangle inequality and the above inequalities)

$$\hat{P}_{S_Q}(h \neq \hat{h}_{S_Q}) \leq C \left( A'_{|S_Q|} \right)^{\frac{\beta_Q}{2 - \beta_Q}}.$$

Thus, in Step 6,

$$c \sqrt{\hat{\delta}(S_Q, S_Q) A_{|S_Q|}} + c A_{|S_Q|} \leq C \left( A'_{|S_Q|} \right)^{\frac{1}{2 - \beta_Q}},$$

which, by our choice of $n_Q^*$ is at most $\epsilon$. Hence, in this case, the algorithm will return in Step 6 (or else would have returned on some previous round).

On the other hand, suppose $S_P$ reaches a size at least $n_P^*$. In this case, again by the same argument used in the proof of Theorem 3, every $h \in \mathcal{H}$ included in the supremum in the definition of $\hat{\delta}(S_P, U_Q)$ has

$$\mathcal{E}_P(h) \leq C \left( A'_{|S_P|} \right)^{\frac{1}{2 - \beta_P}},$$

which implies

$$P(h \neq h^*) \leq C \left( A'_{|S_P|} \right)^{\frac{\beta_P}{2 - \beta_P}},$$

and hence

$$Q(h \neq h^*) \leq C \left( A'_{|S_P|} \right)^{\frac{\beta_P}{(2 - \beta_P)\gamma}}.$$

By the above inequalities and the triangle inequality (since $\hat{h}_{S_P}$ is clearly also included as an $h$ in that supremum), this implies

$$\hat{P}_{U_Q}(h \neq \hat{h}_{S_P}) \leq C \left( A'_{|S_P|} \right)^{\frac{\beta_P}{(2 - \beta_P)\gamma}} + \frac{\epsilon}{8}.$$

Altogether we get that

$$\hat{\delta}(S_P, U_Q) \leq C \left( A'_{|S_P|} \right)^{\frac{\beta_P}{(2 - \beta_P)\gamma}} + \frac{\epsilon}{8}.$$

By our choice of $n_P^*$ (for an appropriate choice of constant factors), the right hand side is at most $\epsilon/4$. Therefore, in this case the algorithm will return in Step 7 (if it had not already returned in some previous round). This completes the proof. $\square$

## E   Proofs for Reweighting Results

The following lemma is known (see [34, 35]), following from the general form of Bernstein's inequality and standard VC arguments, in combination with the well-known fact that, since the VC dimension of $\{(x, y) \mapsto \mathbb{1}[h(x) \neq y] : h \in \mathcal{H}\}$ is $d_{\mathcal{H}}$, and pseudo-dimension of $\mathcal{P}$ is $d_p$, it follows that the pseudo-dimension of $\{(x, y) \mapsto \mathbb{1}[h(x) \neq y] f(x) : h \in \mathcal{H}, f \in \mathcal{P}\}$ is at most $\propto d_{\mathcal{H}} + d_p$.

**Lemma 4.** *With probability at least* $1 - \frac{\delta}{3}$, $\forall f \in \mathcal{P}$, $\forall h, h' \in \mathcal{H}$,

$$R_{P_f}(h) - R_{P_f}(h') \leq \hat{R}_{S_P, f}(h) - \hat{R}_{S_P, f}(h') + c \sqrt{\min\{P_{f^2}(h \neq h'), \hat{P}_{S_P, f^2}(h \neq h')\} A''_{n_P}} + c \|f\|_\infty A''_{n_P}$$

*and* $\frac{1}{2} P_{f^2}(h \neq h') - c \|f\|_\infty A''_{n_P} \leq \hat{P}_{S_P, f^2}(h \neq h') \leq 2 P_{f^2}(h \neq h') + c \|f\|_\infty A''_{n_P}$, *for a universal numerical constant* $c \in (0, \infty)$.

*Proof of Theorem 5.* Let us suppose the event from Lemma 4 holds, as well as the event from Lemma 1 for $S_Q$, and also the part (5) from the event in Lemma 1 holds for $U_Q$. The union bound implies all of these hold simultaneously with probability at least $1 - \delta$. For simplicity, and without loss of generality, we will suppose the constants $c$ in these two lemmas are the same. Regarding the

sufficient size of $|U_Q|$, for this result it suffices to have $|U_Q| \geq n_P^{\frac{\beta_f}{(2-\beta_f)\gamma_f}}$ for all $f \in \mathcal{P}$; for instance, in the typical case where $\gamma_f \geq 1$ for all $f \in \mathcal{P}$, it would suffice to simply have $|U_Q| \geq n_P$.

First note that, exactly as in the proof of Theorem 3, since the event in Lemma 4 implies $h_{P_{\hat{f}}}^*$ satisfies the constraint in the optimization defining $\hat{h}$, and the RCS assumption implies $\mathcal{E}_Q(h_{P_{\hat{f}}}^*) = 0$, and hence by (NC) that $Q(h_{P_{\hat{f}}}^* \neq h_Q^*) = 0$, we immediately get that

$$\mathcal{E}_Q(\hat{h}) \leq C A_{n_Q}^{\frac{1}{2-\beta_Q}}.$$

Thus, it only remains to establish the other term in the minimum as a bound.

Similarly to the proofs above, we let $C_f$ be a general $f$-dependent constant (with the same restrictions on dependences mentioned in the theorem statement), which may be different in each appearance below. For each $f \in \mathcal{P}$, denote by $\hat{h}_f$ the $h \in \mathcal{H}$ that minimizes $\hat{R}_{S_Q}(h)$ among $h \in \mathcal{H}$ subject to $\hat{\mathcal{E}}_{S_P,f}(h) \leq c\sqrt{\hat{P}_{S_P,f^2}(h \neq \hat{h}_{S_P,f})A_{n_P}''} + c\|f\|_\infty A_{n_P}''$. Also note that $\hat{h}_{S_P,f}$ certainly satisfies the constraint in the set defining $\hat{\delta}(S_P, f, U_Q)$, and that the event from Lemma 4 implies $h_{P_f}^*$ also satisfies this same constraint. Therefore, the event for $U_Q$ from Lemma 1, and the triangle inequality, imply

$$\mathcal{E}_Q(\hat{h}_f) \leq Q(\hat{h}_f \neq h_{P_f}^*) \leq Q(\hat{h}_f \neq \hat{h}_{S_P,f}) + Q(h_{P_f}^* \neq \hat{h}_{S_P,f}) \leq 4\hat{\delta}(S_P, f, U_Q) + 4cA_{|U_Q|}.$$

Thus, $\hat{f}$ is being chosen to minimize an upper bound on the excess $Q$-risk of the resulting classifier.

Next we relax this expression to match that in the theorem statement. Again using (5), we get that

$$\hat{\delta}(S_P, f, U_Q) \leq cA_{|U_Q|} +$$
$$2\sup\left\{Q(h \neq \hat{h}_{S_P,f}) : h \in \mathcal{H}, \hat{\mathcal{E}}_{S_P,f} \leq c\sqrt{\hat{P}_{S_P,f^2}(h \neq \hat{h}_{S_P,f})A_{n_P}''} + c\|f\|_\infty A_{n_P}''\right\}.$$

Again since $h_{P_f}^*$ and $\hat{h}_{S_P,f}$ both satisfy the constraint in this set, the supremum on the right hand side is at most

$$2\sup\left\{Q(h \neq h_{P_f}^*) : h \in \mathcal{H}, \hat{\mathcal{E}}_{S_P,f} \leq c\sqrt{\hat{P}_{S_P,f^2}(h \neq \hat{h}_{S_P,f})A_{n_P}''} + c\|f\|_\infty A_{n_P}''\right\}.$$

Then using the marginal transfer condition, this is at most

$$C_f\sup\left\{P_{f^2}(h \neq h_{P_f}^*)^{\frac{1}{\gamma_f}} : h \in \mathcal{H}, \hat{\mathcal{E}}_{S_P,f} \leq c\sqrt{\hat{P}_{S_P,f^2}(h \neq \hat{h}_{S_P,f})A_{n_P}''} + c\|f\|_\infty A_{n_P}''\right\},$$

and the Bernstein Class condition further bounds this as

$$C_f\sup\left\{\mathcal{E}_{P_f}^{\frac{\beta_f}{\gamma_f}} : h \in \mathcal{H}, \hat{\mathcal{E}}_{S_P,f} \leq c\sqrt{\hat{P}_{S_P,f^2}(h \neq \hat{h}_{S_P,f})A_{n_P}''} + c\|f\|_\infty A_{n_P}''\right\}.$$

Finally, by essentially the same argument as in the proof of Theorem 3 above, every $h \in \mathcal{H}$ with $\hat{\mathcal{E}}_{S_P,f} \leq c\sqrt{\hat{P}_{S_P,f^2}(h \neq \hat{h}_{S_P,f})A_{n_P}''} + c\|f\|_\infty A_{n_P}''$ satisfies

$$\mathcal{E}_{P_f}(h) \leq C_f(A_{n_P}'')^{\frac{1}{2-\beta_f}},$$

so that the above supremum is at most $C_f(A_{n_P}'')^{\frac{\beta_f}{(2-\beta_f)\gamma_f}}$ for a (different) appropriate choice of $C_f$. Altogether we have established that

$$\hat{\delta}(S_P, f, U_Q) \leq cA_{|U_Q|} + C_f(A_{n_P}'')^{\frac{\beta_f}{(2-\beta_f)\gamma_f}}.$$

By our condition on $|U_Q|$ specified above, this implies

$$\hat{\delta}(S_P, f, U_Q) \leq C_f(A_{n_P}'')^{\frac{\beta_f}{(2-\beta_f)\gamma_f}}.$$

We therefore have that

$$\mathcal{E}_Q(\hat{h}) = \mathcal{E}_Q(\hat{h}_{\hat{f}}) \leq 4\hat{\delta}(S_P, \hat{f}, U_Q) + 4cA_{|U_Q|} = \inf_{f \in \mathcal{P}} 4\hat{\delta}(S_P, \hat{f}, U_Q) + 4cA_{|U_Q|}$$

$$\leq \inf_{f \in \mathcal{P}} C_f(A''_{n_P})^{\frac{\beta_f}{(2-\beta_f)\gamma_f}},$$

where we have again used the condition on $|U_Q|$. This completes the proof. $\square$