[Reviews · NeurIPS 2019]

Reviewer 1



Originality: I think the analysis given is novel, and related work is sufficiently cited and contrasted to the proposed discrepancies. What is especially nice is the section with examples that shows the benefit of the proposed discrepancies, and the authors even go to the length of trying to fix the $\gamma$ discrepancy, but show that even this fixed version may not be competitive with their proposed measure. I also think that the distributional conditions are novel and not usually introduced in theoretical analyses I've seen before. It also has many contributions. Regarding citations: - [6] has recently been published in JMLR, please update the citation - line 157, please add the Wasserstein notion to the related work Quality: - The work is extremely technical, and sadly, I could not check the proofs - The work is purely theoretical with no practical component. The work seems completely finished, and all proofs are there in the supplement. There are some typo's but nothing major, I think the work was well proofread. 3) One major issue I find is: the authors do not reflect on their own work, its limits, or its applicability. There is no Discussion, Conclusion or Future Work section. I think this is majorly important and should be added. 4) In particular, for a NIPS audience that is a bit more interested in practical algorithms, it would be nice if the authors reflect on whether the proposed algorithms and theory are practically applicable. I'm not asking for a practical algorithm (which is clearly out of the scope of this theoretical paper which already has enough contributions), but at least the authors could say something about what steps are necessary to bring this theory into practice. For example, are the proposed algorithms computeable, are the optimizations do-able (convex?) or perhaps are they NP-hard? What is the complexity? And for example, when we need distributions f (for the reweighting algorithm), how would we go about choosing them? I think the paper would be much more influential if the authors could say something about this (even if only one or two paragraphs), because it could be helpful for non-theory people to try and build practical algorithms based on the insights of this work 5) I don't see how the d_A and d_\Gamma discrepancies imply any rates (I have not seen those in the literature). Can you back this up with an explanation or citation? 6) Line 34. If so I understand that for the hypothesis-transfer approach, not just simply \hat{h}_P or \hat{h}_Q is selected? This was a bit unclear for my first reading. 7) line 245: Why 'for our purposes'? This is also vague. Is this choice necessary? 8) line 253: well chosen in what way? I find this a bit vague.... 9) line 251: what is near-minimal? why? can you elaborate on this 10) line 266: could you say something about the choice of the set P? could we choose a set of parametric functions or non-parametric ones? do they need support on all of X or could it be sufficient to choose the function to only have support on the observed samples S_P and S_Q? For these clarifications, maybe they could be added to the supplement. Clarity: I think the work is extremely dense and difficult to read... It took me a long time, and I had to go back and forth a lot. I think the clarity can be greatly improved. After reading it, and going back to the introduction, I also could not match all content with all things mentioned in the Introduction. Either some things or arguments are missing, or they were not explicitly in the main body, or it was too unclear to match up everything (which, after spending so much time on reading this paper, you would hope I would be able to do). I will now give some suggestions to improve the clarity: 11) In the introduction, give a list of contributions, and in which sections we can find them. Try to go over all sections, then I don't need to figure out what is where by myself, and I know in what order I can expect the material to come. It would also be nice if you mention which theorems and algorithms are where, so if needed, you can refer to them in the main text before they are introduced. Such as in line 136, you refer to a Theorem that has yet to come, and in line 202: Here I would find it useful to give a reference to the Theorem that does that (the best upper bound). 12) Introduction: I think it would be good to give a motivation and explenation for a broader audience about what is the problem. Why do we care about transfer learning in the first place? And possibly give a short explanation: what is transfer learning. You could also refer to a survey paper for readers that are interested. 13) I think also, if you give a Discussion and / or Conclusion section, you could summarize and point back to the work, reiterate what are the main points. I was missing this. 14) after reading the paper and going back to the introductions, I feel some things are missing: 14.1) line 30: I did not see, where it was shown that any classifier that has access to both P and Q, would not achieve the optimal Q rate. maybe this can be made more explicit 14.2) line 31: it would be good to give an argument in the corresponding section that explains the argument that cross-validating naively gives a suboptimal rate, and why. 14.3) line 33: what is hypothesis transfer? reference / citation? 14.4) line 36: I found it a bit confusing, that the marginal transfer-component actually depends on h^*. so it actually does not only depend on the marginal, but also indirectly on P(Y|X) through h^*. maybe its good to clarify that or make it a bit more explicit. 14.5) line 38: after reading the paper, it is still a bit unclear to me why we need the marginal exponent $\gamma$ instead of $\rho$ for the case of unlabeled data. since for both $\rho$ and $\gamma$ depend on P(Y|X) through h_P^*. or is it because in the case of unlabeled data, we can estimate h_P^*, and therefore $\gamma$ is more appliceable than $\rho$, since $\rho$ depends on the entire $P(X,Y)$? 14.6) line 33, the generic approach is first introduced (Section 5) and then the oracle algorithm that chooses either h_P or h_Q is introduced (Section 6). but this does not match the order of the introduction. or it may be the case Im misstaken and this is not Section 6, then I simply not could not find back this argument: that an oracle that ignores data can be optimal (but how to choose h_Q or h_P). 15) in Line 68 'However, a significant downside to these notions ...' what does these notions refer to? The proposed transfer-exponent notions, or the KL-difergence and Renyi divergence? More detailed comments: 16) Line 36: The sentence is not grammatically correct. Also, I do not understand what it means that 'it most accurately capture the fact that practical decisions in transfer most often involve'... means. I think this sentence is very convoluted... I would propose: We then propose a related notion of marginal transfer-exponent, which computes discrepancy w.r.t. the marginals, which can use unlabeled data. This is of practical relevance since unlabeled data is often cheaper to collect.' 17) Line 17: generally, in most domain adaptation and transfer learning papers I've read, I think the accepted practice is to indicate the source with Q and the target with P. 18) Line 28: I think there may be a typo. What is ignoring the 'worse' of the data? 19) line 175: I find the section title contains very little information. Maybe 'Lower-Bounds for Transfer' 20) line 208: I would like to see the section title improved here also 21) line 231: I would include in the section title 'unlabeled data' to improve the clarity 22) line 97: replace 'Transfer' with 'transfer learning'. 23) line 91, I think it would be nice to explain that NC stands for Noise Condition and RCS stands for Relaxed Covariate Shift 24) line 233: 'The idea is that ... in many applications'. This sentence is a bit empty 'X is true, so X is realistic'. Significance: I think the work is very important. It provides new insights into the problem of transfer and what the limits are of the setting: e.g. what kind of improvements are possible. This will definitely be important for the community. Also, several algorithms are given that may be of practical interest. In particular, the non-symmetric transfer component is interesting, and the example of 'super-transfer' is also surprising and cool. I think generally, the authors may have a problem with the pagelimit... But really, the proposed additions (contribution list, discussion / futurework, conclusion) could greatly benefit the paper. Your readers will really appreciate it! My suggestion is maybe to place some of the algorithms towards the supplement, to make space for these important additions. For example, you could easily push Section 7.1 to the supplement. Update after author feedback: - Seeing as the authors will move the proof ideas to the supplement, and since the authors at least have addressed 3 (discussion / limitations) 4 (practical / algorithmic questions), 13 (summarizing and listing contributions), I thrust 12 and 14 will be sufficiently addressed in the camera ready version. Thus I have changed my score to 9 as promised. Great paper and good luck preparing the camera ready version, looking forward to read it.

Reviewer 2



Originality and significance: It is clear that this work makes substantial contributions to understanding worst-case behavior of transfer learning. Clarity: Given the heavily theoretical nature of the work, the submission's clarity can be improved in a few ways. A few comments on writing and structure: - From reading the introduction, it is unclear what the marginal transfer exponent is, and why it's needed. - Clearly state that the paper only studies binary classification problems with 0-1 losses. From the first two pages, it is unclear that the paper only deals with purely theoretical topics. This is not to say the paper adds meaningful contributions to learning theory. Introduction is somewhat misleading, and makes it hard to place the true contributions (and limitations) of this paper. - Throughout the paper, the use of "\gamma" and "\rho" as pronouns for transfer exponents is confusing, as these are not uniquely defined concepts. (e.g. Line 123 or Example 3) - Examples 2-4 would be better served if presented after Theorem 3. The forward referencing is confusing at best. - A brief explanation of why Theorem 2 is stated in terms of expectations would be helpful. - I'm guessing \epsilon_U = \epsilon_H in Remark 1. - Line 179: Define the VC dimension notation first before using it. - Line 57: "first first", Line 70: "does the", Line 74: "efforst", Line 200: second \epsilon_1 should be \epsilon_2

Reviewer 3



This paper aims to contribute to the theory of transfer learning by making new divergence measures across domains to develop minimax bounds, bounds on sampling cost and covariate shift correction bounds using instance reweighting. While it is very useful to have minimax theory on the transfer learning setting (in this case covariate shift), it is difficult to parse significance of the theoretical results in the way the paper is written. In particular, the current results lack sufficient comparison with prior work, for example with transductive bounds for covariate shift, such as in [Cortest et al, 2008, Gretton et al, 2009]. Furthermore, there does not appear to be sufficient treatment of the results as to how they depend on various terms. For example, there is almost no discussion on the implications of Theorem 1, 3, 4, 5 and 6. However, the new quantities introduced by the authors to measure divergence, and the kinds of bounds being derived can be useful if they are put into better context, and more intuition is provided about the theoretical results presented References: Cortes, Corinna, et al. "Sample selection bias correction theory." International conference on algorithmic learning theory. Springer, Berlin, Heidelberg, 2008. Gretton, Arthur, et al. "Covariate shift by kernel mean matching." Dataset shift in machine learning 3.4 (2009): 5.

Reviewer 4



* The addressed problem is of theoretical and practical interest in transfer learning. Starting from the definition of transfer exponents, a notion introduced in [1], the paper proposes several theoretical results on bounding the excess risk over the target domain under different settings (sampling cost minimization, source data re-weighting, choosing from multiple sources). These results are established for any classifier from a class of functions with some VC dimension. My feeling is that the assumptions and the obtained results are meaningful and reasonable, at least from conceptual point of view of transfer learning benefits. * All results heavily rely on the relaxation of covariate shift assumption. Do the established results still hold if the covariate assumption is strictly enforced that is P_Y|X = Q_Y|X. If not, how the rates change? * The related work to this submission is the paper of Kpotufe and Martinet [1]. How the obtained results contrast with the ones presented in [1]? Discussion and analysis on this point may be of interest to the reader. * Usually, sampling labels for target data may be much costly than for source data. From Thm 4, what are the transition regimes for $n_p$ and $n_Q$ w.r.t. sampling costs $c_P$ and $c_Q$? Under which condition sampling few target labels is sufficient to attain the desired excess risk. * All learning procedures proposed in the paper rely on probabilities $P_D(h \neg h')$. How these probabilities are evaluated in practice without explicit knowledge of source and target distributions? To turn these procedures into practical algorithms, an instantiation on a family of classifiers (linear model, k-NN or others) may be interesting and helpful. Minor comments ------------------- * Lines 141-145: the text is hard to read. * Line 249 (learning procedure): parameter $\tilde C$ is defined at line 0 and is not applied afterwards. Lines 2 and 4: parameter $C$ is undefined. Line 7: can the paper provides an intuition about the upper bound $\epsilon/4$ used to select hypothesis $\hat h_{S_P}$?

[Author Response · NeurIPS 2019]

**Clarity and Positioning.** While we were somewhat surprised about the comments in this regard (we dedicated 3-4 pages to just discussions and positioning, including nearly a page of related work), we now believe after careful considerations of the reviews that some of these discussions did not appear in the places where readers might expect them, e.g., closer to theorem statements. Given the page limits we had instead opted to discuss proof ideas close to theorem statements but should have reiterated and referenced previous discussions. We will move proof ideas to the appendix, and heed the various suggestions. In terms of related work, we strived to diligently overview the vast literature, although we missed some as pointed out by one reviewer (Reviewer 3); we however emphasize that the suggested references, while relevant, are somewhat peripheral to our main contributions (see below).

**Significance and Contributions.** Our main contributions are twofold:
(1) We derive the *first* general measures of discrepancy (transfer exponents) **that yield tight rates in transfer** for convergence of excess risk to zero. We argue that these new measures overcome significant limitations suffered by other discrepancies previously proposed in the literature, thereby describing many scenarios where transfer is possible but where this fact is not implied by any previously-defined notions of discrepancy ($d_\mathcal{A}$, $L_p$, KL-divergence, Kernel-mean discrepancy, Wasserstein). This is due e.g. to the fact that transfer is inherently an *asymmetric problem* (one distribution might have sufficient information on the other but not the other way around), so cannot be captured by metrics ($L_1$, $d_\mathcal{A}$, Wasserstein, Kernel-mean discrepancy), and does not require that distributions have the same support (KL-divergence, density ratios). The recent notion of [18] has a similar flavor, but is concerned with specific smoothness assumptions on the regression function $\mathbb{E}[Y|X]$ and does not have a way to take into account the structure of an abstract hypothesis class $\mathcal{H}$: for instance, [18] still requires distributions with the same support (as discussed in Related work section, and Example 1, lines 77-80 and 124-131). Indeed, our starting point for this work was to try to extend the insights of [18] to yield a general $\mathcal{H}$-dependent discrepancy measure (for the purpose of overcoming shortcomings in the well-known $d_\mathcal{A}$ and $d_\mathcal{Y}$ discrepancies, as we discuss in Example 2).

(2) We show for the first time that many questions of a practical interest can in principle be addressed in a near-optimal way, *without estimating the discrepancy between distributions*, but through clever use of unlabeled data. For instance, nearly all previous works on transfer assumed no labeled target data, and therefore could not address practical questions such as the optimal mix of source and target labeled data (of potentially different sampling costs); in this regard, we give the first bounds **in terms of any mix of $n_P$ source and $n_Q$ target labeled samples for VC classes** and show that these bounds are tight. What's surprising here is that the optimal rate behaves as $\min\{\mathcal{E}_Q(\hat{h}_P), \mathcal{E}_Q(\hat{h}_Q)\}$: i.e., is akin to ignoring one of the source or target data sets (but note that we still need to use both samples in order to decide which one to ignore in the end). This is then essential in assessing optimal sampling under mixed costs.

We *mitigate the usual worst-case nature of minimax analysis* by showing that our bounds are **tight for any given hypothesis class**, and, **tight in any noise regime** (Theorems 1 and 2). We are not aware of any previous such individual tightness result in the context of transfer.

**What we left open.** In this first step, we mainly focused on understanding the theoretical limits of transfer under mixed sampling costs, and outlined various theoretical procedures (non-implementable as they are based on ERM on 0-1 loss), which yield the insights that various practical questions can be addressed in principle with no prior distributional knowledge: i.e., from data-driven decisions alone. Given the simplicity of the algorithmic principles outlined, a natural next step is to investigate practical versions of these procedures based on convex surrogate losses, and extensions to address multiclass learning, regression, and other general prediction settings.

**Other questions by reviewers.** Space limits prevent answering all reviewer questions, but we address a few below.
*Reviewers 1 and 2.* The marginal transfer exponent $\gamma$ indeed depends on unknown $h_P^*$. However we never have to estimate it, as we show that the ERM $\hat{h}_P$ on source data is a good surrogate for $h_P^*$. Hence, using *cheap* unlabeled data, we can identify good hypotheses $h$ using the fact that $Q_X(h \neq \hat{h}_P) \approx Q_X(h \neq h_P^*)$ which upper-bounds the excess error $\mathcal{E}_Q(h)$ (this is all done rigorously in the paper). Unfortunately, using unlabeled data to drive choices, all we can adapt to is the marginal exponent which, unlike $\rho$, requires little knowledge of $Q_{Y|X}$. Note however that if all we are interested in is the basic transfer problem of using $n_P$ and $n_Q$ given samples (sections 4 and 5), then we can adapt optimally to even $\rho$ without estimating it (Theorem 3), which is surprising as we know very little about $Q_{Y|X}$ when $n_Q$ is small. Regarding the question about $\mathcal{P}$, any set defined a-priori is fine (not data-dependent).

*Reviewer 3.* We hope many of your concerns were addressed above. The references mentioned will be added. They use metric notions of discrepancy (see above), and in contrast to our work, largely leave open the question of tightness.

*Reviewer 4.* The results allow $P_{Y|X} = Q_{Y|X}$: i.e., we do not *require* them to be different in the upper bounds (and there exist classes where the lower bounds hold even with restriction to $P_{Y|X} = Q_{Y|X}$). As per Theorem 4, few target labels are required whenever $\gamma$ is small and $\mathfrak{c}_P \ll \mathfrak{c}_Q$; we will add relevant comments to the paper. The results hold for *any* VC class, and are instantiated simply by plugging in the VC dimension of the class.
We thank you for pointing out the typo in line 249. "$C$" is meant to be $\tilde{C}$ and in fact should double at each iteration.

[Meta-Review · NeurIPS 2019]

This is a solid paper on the generalization theory of transfer learning. To ease the readability and extract the main messages of the paper, the authors should take time to better discuss the technical concepts they manipulate.